# MOVING FORWARD BY MOVING BACKWARD: EMBEDDING ACTION IMPACT OVER ACTION SEMANTICS

**Kuo-Hao Zeng**[1]**, Luca Weihs**[2]**, Roozbeh Mottaghi**[1]**, Ali Farhadi**[1]
[1]Paul G. Allen School of Computer Science & Engineering, University of Washington
[2]PRIOR @ Allen Institute for AI
prior.allenai.org/projects/action-adaptive-policy

## ABSTRACT

A common assumption when training embodied agents is that the impact of taking an action is stable; for instance, executing the "move ahead" action will always move the agent forward by a fixed distance, perhaps with some small amount of actuator-induced noise. This assumption is limiting; an agent may encounter settings that dramatically alter the impact of actions: a move ahead action on a wet floor may send the agent twice as far as it expects and using the same action with a broken wheel might transform the expected translation into a rotation. Instead of relying that the impact of an action stably reflects its pre-defined semantic meaning, we propose to model the impact of actions on-the-fly using latent embeddings. By combining these latent action embeddings with a novel, transformer-based, policy head, we design an Action Adaptive Policy (AAP). We evaluate our AAP on two challenging visual navigation tasks in the AI2-THOR and Habitat environments and show that our AAP is highly performant even when faced, at inference-time with missing actions and, previously unseen, perturbed action space. Moreover, we observe significant improvement in robustness against these actions when evaluating in real-world scenarios.

## 1 INTRODUCTION

Humans show a remarkable capacity for planning when faced with substantially constrained or augmented means by which they may interact with their environment. For instance, a human who begins to walk on ice will readily shorten their stride to prevent slipping. Likewise, a human will spare little mental effort in deciding to exert more force to lift their hand when it is weighed down by groceries. Even in these mundane tasks, we see that the effect of a humans' actions can have significantly different outcomes depending on the setting: there is no predefined one-to-one mapping between actions and their impact. The same is true for embodied agents where something as simple as attempting to moving forward can result in radically different outcomes depending on the load the agent carries, the presence of surface debris, and the maintenance level of the agent's actuators (*e.g.*, are any wheels broken?). Despite this, many existing tasks designed in the embodied AI community (Jain et al., 2019; Shridhar et al., 2020; Chen et al., 2020; Ku et al., 2020; Hall et al., 2020; Wani et al., 2020; Deitke et al., 2020; Batra et al., 2020a; Szot et al., 2021; Ehsani et al., 2021; Zeng et al., 2021; Li et al., 2021; Weihs et al., 2021; Gan et al., 2021; 2022; Padmakumar et al., 2022) make the simplifying assumption that, except for some minor actuator noise, the impact of taking a particular discrete action is functionally the same across trials. We call this the *action-stability assumption* (AS assumption). Artificial agents trained assuming action-stability are generally brittle, obtaining significantly worse performance, when this assumption is violated at inference time (Chattopadhyay et al., 2021); unlike humans, these agents cannot adapt their behavior without additional training.

In this work, we study how to design a reinforcement learning (RL) policy that allows an agent to adapt to significant changes in the impact of its actions at inference time. Unlike work in training robust policies via domain randomization, which generally leads to learning conservative strategies (Kumar et al., 2021), we want our agent to fully exploit the actions it has available: philosophically, if a move ahead action now moves the agent twice as fast, our goal is not to have the agent take smaller steps to compensate but, instead, to reach the goal in half the time. While prior works have studied test time adaptation of RL agents (Nagabandi et al., 2018; Wortsman et al., 2019; Yu et al., 2020; Kumar et al., 2021), the primary insight in this work is an action-centric approach which

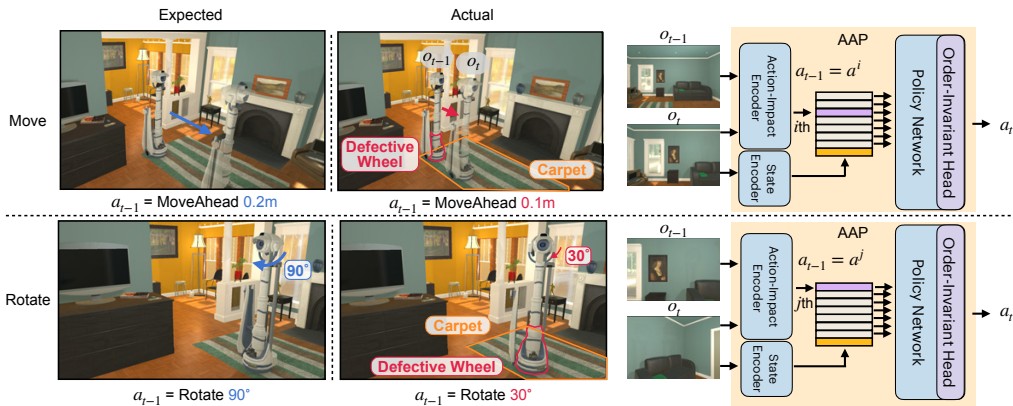

Figure 1: An agent may encounter unexpected drifts during deployment due to changes in its internal state (*e.g.*, a defective wheel) or environment (*e.g.*, hardwood floor *v.s.* carpet). Our proposed Action Adaptive Policy (AAP) introduces an action-impact encoder which takes state-changes (e.g., $o_t \rightarrow o_{t+1}$) caused by agent actions (e.g., $a_{t-1}$) as input and produces embeddings representing these actions' impact. Using these action embeddings, the AAP utilizes a Order-Invariant (OI) head to choose the action whose impact will allow it to most readily achieve its goal.

requires the agent to generate action embeddings from observations on-the-fly (i.e., no pre-defined association between actions and their effect) where these embeddings can then be used to inform future action choices.

In our approach, summarized in Fig. 1, an agent begins each episode with a set of unlabelled actions $\mathcal{A} = \{a^0, ..., a^n\}$. Only when the agent takes one of these unlabelled actions $a^i$ at time $t$, does it observe, via its sensor readings, how that action changes the agent's state and the environment. Through the use of a recurrent action-impact encoder module, the agent then embeds the observations from just before ($o_t$) and just after ($o_{t+1}$) taking the action to produce an embedding of the action $e_{i,t}$. At a future time step $t'$, the agent may then use these action-impact embeddings to choose which action it wishes to execute. In our initial experiments, we found that standard RL policy heads, which generally consist of linear function acting on the agent's recurrent belief vector $b_t$, failed to use these action embeddings to their full potential. As we describe further in Sec. 4.3, we conjecture that this is because matrix multiplications impose an explicit ordering on their inputs so that any linear-based actor-critic head must treat each of the $n!$ potential action orderings separately. To this end, we introduce a novel, transformer-based, policy head which we call the Order-Invariant (OI) head. As suggested by the name, this OI head is invariant to the order of its inputs and processes the agent's belief jointly with the action embeddings to allow the agent to choose the action whose impact will allow it to most readily achieve its goal. We call this above architecture, which uses our recurrent action-impact encoder module with our OI head, the Action Adaptive Policy (AAP).

To evaluate AAP, we train agents to complete two challenging visual navigation tasks within the AI2-THOR environment (Kolve et al., 2017): Point Navigation (PointNav) (Deitke et al., 2020) and Object Navigation (ObjectNav) (Deitke et al., 2020)[1]. For these tasks, we train models with moderate amounts of simulated actuator noise and, during evaluation, test with a range of modest to severe unseen action impacts. These include disabling actions, changing movement magnitudes, rotation degrees, *etc.*; we call these action augmentations *drifts*. We find that, even when compared to sophisticated baselines, including meta-learning (Wortsman et al., 2019), a model-based approach (Zeng et al., 2021), and RMA (Kumar et al., 2021), our AAP approach handily outperforms competing baselines and can even succeed when faced with extreme changes in the effect of actions. Further analysis shows that our agent learns to execute desirable behavior at inference time; for instance, it quickly avoids using disabled actions more than once during an episode despite not being exposed to disabled actions during training. In addition, the experimental results in a real-world test scene from RoboTHOR on the Object Navigation task demonstrate that our AAP performs better than baselines against unseen drifts.

In summary, our contributions include: (1) an action-centric perspective towards test-time adaptation, (2) an Action Adaptive Policy network consisting of an action-impact encoder module and a novel order-invariant policy head, and (3) extensive experimentation showing that our proposed approach outperforms existing adaptive methods.

---

[1]We also show results in a modified PettingZoo environment Terry et al. (2020) on Point Navigation and Object Push in Sec. F and a modified Habitat Environment (Savva et al., 2019) on Point Navigation in Sec. G.

## 2  RELATED WORK

**Adaptation.**  There are various approaches in the literature that address adaptation to unknown environments, action effects, or tasks.

*Novel environments:* These approaches explore the adaptation of embodied agents to unseen environment dynamics (Yu et al., 2019; Wortsman et al., 2019; Zhou et al., 2019; Li et al., 2020a;b; Peng et al., 2020; Song et al., 2020; Loquercio et al., 2021; Evans et al., 2022; O'Connell et al., 2022; Kumar et al., 2021; 2022; Zhang et al., 2022; Agarwal et al., 2022). Various techniques such as meta-learning  (Wortsman et al., 2019), domain randomization (Yu et al., 2019), and image translation (Li et al., 2020a), have been used for adaptation. In contrast to these approaches, we address changes in the actions of the agent as well. Moreover, unlike some of these approaches, *e.g.* Li et al. (2020a), we do not assume access to the test environment.

*Damaged body and perturbed action space:* These methods focus on scenarios that the effect of the actions changes during inference time as a result of damage, weakened motors, variable load, or other factors. Yu et al. (2020) study adaptation to a weakened motor. Nagabandi et al. (2018) explore adaptation to a missing leg. Yu et al. (2017) adapt to differences in mass and inertia of the robot components. Our approach is closer to those of Yu et al. (2020) and Nagabandi et al. (2018) that also consider changes in environment structures as well. Nonetheless, these works focus on using meta-learning for locomotion tasks and optimize the model at the testing phase while we do not optimize the policy at inference time. In our experiments, we find we outperform meta-learning approaches without requiring, computationally taxing, meta-learning training.

*Novel tasks:* Several works focus on adapting to novel tasks from previously learned tasks or skills (Finn et al., 2017; Gupta et al., 2018; Chitnis et al., 2019; Huang et al., 2019; Fang et al., 2021). We differ from these methods as we focus on a single task across training and testing.

**Out-of-distribution generalization.** Generalization to out-of-distribution test data has been studied in other domains such as computer vision (Hendrycks & Dietterich, 2019; Peng et al., 2019), NLP (McCoy et al., 2019; Zhang et al., 2019), and vision & language (Agrawal et al., 2018; Akula et al., 2021). In this paper, our focus is on visual navigation, which in contrast to the mentioned domains, is an interactive task and requires reasoning over a long sequence of images and actions.

**System identification.**  Our paper shares similarities with the concept of System Identification (Verma et al., 2004; Bongard et al., 2006; Seegmiller et al., 2013; Cully et al., 2015; Banerjee et al., 2020; Lew et al., 2022). The major difference between our approach and the mentioned works is that we use visual perception for adaptation.

## 3  PROBLEM FORMULATION

In this work, we aim to develop an agent which is robust to violations of the AS assumption. In particular, we wish to design an agent that quickly adapts to settings where the outcomes of actions at test time differ, perhaps significantly, from the outcomes observed at training time; for instance, a rotation action might rotate an agent twice as far as it did during training or, in more extreme settings, may be disabled entirely. We call these unexpected outcomes of actions, *drifts*. As discussed in Sec. 1, the AS assumption is prevalent in existing embodied AI tasks and thus, to evaluate how well existing models adapt to potential drifts, we must design our own evaluation protocol. To this end, we focus on two visual navigation tasks, Point and Object Navigation (PointNav and ObjectNav), as (1) visual navigation is a well-studied problem with many performant, drift-free or fixed-drift, baselines, and (2) the parsimonious action space used in visual navigation (Move, Rotate, and End) allows us to more fully explore the range of drifts possible for these tasks. We will now describe the details of how we employ drifts to evaluate agents for these tasks.

In this work, we assume that a particular drift, perhaps caused by a defective sensor, broken parts, motor malfunction, different amount of load, or stretched cable (Boots et al., 2014), may change across episodes. We focus primarily on two categories of potential drift occurring in visual navigation tasks: movement drift $d_m$, which causes an unexpected translation when executing a Move action, and rotation drift $d_r$, which leads to an unexpected rotation when executing a Rotate action. More specifically, we semantically define the movement and rotation actions as Move($d$) = "move forward by $d$ meters" and Rotate($\theta$) = "rotate by $\theta$ degrees clockwise". As we describe below, the semantic meaning of these actions is designed to be true, up to small-to-moderate noise, during training, but may change significantly during evaluation.

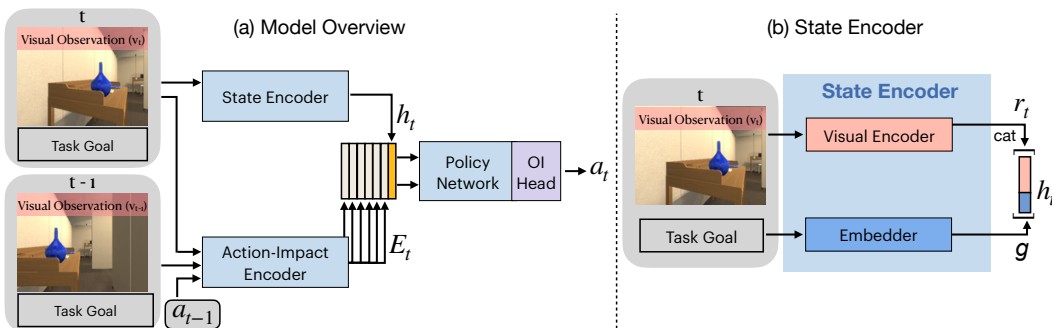

Figure 2: **(a) Model Overview.** Our model includes three modules: a state encoder, an action-impact encoder, and a policy network with an order-invariant head (OI head). The blue and purple colors denote learnable modules, and the yellow and light gray color represents hidden state from the state encoder and action embedding from the action-impact encoder. **(b) State Encoder** is composed of a visual encoder for visual encoding and a embedder for task goal. The light red and dark blue colors indicate the learnable visual encoder and goal embedder, respectively.

For each training and evaluation episode, a movement drift $d_m$ and a rotation drift $d_r$ are sampled and fixed throughout the episode. At every step, upon executing `Move(d)` the agent experiences a $d + d_m + n_d$ translation toward its forward-facing direction, where the $n_d$ represents the high-frequency actuator noise from the RoboTHOR environment (Deitke et al., 2020). Similarly, when the agent executes `Rotate(θ)`, the agent rotates by $θ + d_r + n_θ$ degrees where $n_θ$ again represents high-frequency actuator noise. To evaluate robustness to the AS assumption, the strategies we use to choose $d_m$ and $d_r$ differ between training and evaluation episodes.

**During training.** At the start of a training episode, we sample movement drift $d_m \sim U(-p, p)$ and rotation drift $d_r \sim U(-q, q)$, where $U(\cdot, \cdot)$ denotes a uniform distribution and $|p| \ll u_m$ as well as $|q| \ll 180°$. We set $u_m = 0.25$m, the standard movement magnitude in RoboTHOR.

**During inference.** At the start of an evaluation episode, we sample movement drift $d_m \notin [-p, p]$ and a rotation drift $d_r \notin [-q, q]$.

Note that the drifts chosen during evaluation are disjoint from those seen during training.

## 4 ACTION ADAPTIVE POLICY

We first overview our proposed Action Adaptive Policy (AAP) in Sec. 4.1. We then present the details of our action-impact encoder and Order-Invariant (OI) head in Sec. 4.2 and Sec. 4.3, respectively. Finally, we describe our training strategy in Sec. 4.4.

### 4.1 MODEL OVERVIEW

The goal of our model is to adapt based on action impacts it experiences at test time. To accomplish this, the Action Adaptive Policy (AAP) includes three modules: a state encoder, an action-impact encoder, and a policy network with an order-invariant head (OI head), as illustrated in Fig. 2 (a). The state encoder is responsible for encoding the agent's current observation, including a visual observation $v_t$ and the task goal, into a hidden state $h_t$. The action-impact encoder processes the current observation, previous observation, and previous action $a_{t-1}$ to produce a set of action embeddings $E_t$, one embedding for each action in the action space $\mathcal{A}$. More specifically, the job of the action-impact encoder is to update the action embeddings via the previous action recurrently using the encoded *state-change* feature (Sec. 4.2). Note that the embeddings $e_{i,t} \in E_t$ are not given information regarding which action was called to produce them; this is done precisely so that the agent cannot memorize the semantic meaning of actions during training and, instead, must explicitly model the impact of each action. Finally, the policy network with the OI head takes both the hidden state $h_t$ and the set of action embeddings $E_t$ as inputs and predicts an action $a_t$.

**State Encoder.** Following Batra et al. (2020b); Weihs et al. (2020); Khandelwal et al. (2022), we implement the state encoder with a visual encoder and an embedder, as shown in Fig 2 (b). Note that popular baseline policy models used for embodied AI tasks frequently employ a policy network directly after the state encoder to produce the agent's action. In this work, we use an RGB image as our visual observation $v_t$. For the *PointNav* task goal, we follow Savva et al. (2019) and use both GPS and compass sensors to provide the agent with the angle and distance to the goal position, $\{\Delta\rho_t, \Delta\phi_t\}$. For the *ObjectNav* task, we follow Deitke et al. (2020) and provide the agent with a semantic token corresponding to the goal object category as the task goal. We use a CLIP-pretrained

Figure 3: **Action-Impact Encoder.** The input to the action-impact encoder are two consecutive observations and the previous action. The encoder first extracts visual representations and a goal representation via a ResNet-50 and an Embedder. Concatenated, these form a state-change feature $f_t$. The encoder then uses the previous action $a_{t-1}=a^i$ to retrieve the corresponding memory $m_i$. With $m_i$, an RNN maps $f_t$ to an embedding. Finally, the encoder registers this embedding as the action embedding $e_{i,t}$ if $a^i$ is not a "special" action (*i.e.*, a non-actuator-based action). Otherwise, the encoder registers an action embedding obtained from the Action-Order Encoder into $e_{i,t}$.

ResNet-50 (He et al., 2016; Radford et al., 2021) as our visual encoder and a multi-layer perceptron (MLP) as the embedder to obtain visual representation $r_t$ and goal representation $g$, respectively. Finally, we concatenate them to construct the hidden state $h_t$.

## 4.2 ACTION-IMPACT ENCODER

In this section, we introduce the action-impact encoder, see Fig. 3. The goal of this encoder is to produce a set of action embeddings $E_t$ which summarize the impact of actions the agent has taken in the episode so far. To adapt to unseen drifts during inference, these action embeddings should not overfit to action semantics, instead they should encode the impact that actions actually have at inference time. The action-impact encoder first extracts a state-change feature $f$ from two consecutive observations. It then utilizes a recurrent neural network (RNN) to update the action embedding $e_{i,t}$ according to the previous action $a_{t-1}=a^i$. As the action-impact encoder generates embeddings for each action $a^i$ without using semantic information about the action, every embedding $e_{i,t}$ only encodes the effect of its corresponding action. The decision to not use information about action semantics has one notable side-effect: at the start of an episode, all action embeddings $e_{i,0}$ will be equal so that the agent will not know what any of its actions accomplish. It is only by using its actions, and encoding their impact, that the agent can learn how its actions influence the environment.

During the feature extraction stage, we employ the same visual encoder and embedder used by the state encoder to process the visual observations, $v_t$ and $v_{t-1}$, and the task goal. A linear layer is used to project the two concatenated visual representations $[r_t, r_{t-1}] \in \mathbb{R}^l$ to $r_t^* \in \mathbb{R}^{\frac{l}{2}}$, where $l = 1024$ for *PointNav* and $l = 3136$ for *ObjectNav*. We then concatenate $r_t^*$ and the goal representation $g$, to form the state-change feature $f_t = [r_t^*, g]$, see the *Feature Extract* stage of Fig. 3.

After feature extraction, we apply an RNN, namely a GRU (Chung et al., 2014), to summarize state changes through time, as illustrated in the *Encode* stage of Fig. 3. The use of a recurrent network here allows the agent to refine its action embeddings using many observations through time and also ignore misleading outliers (*e.g.*, without a recurrent mechansim, an agent that takes a `Move` action in front of a wall may erroneously believe that the move action does nothing). To update the action embeddings recurrently, we utilize the previous action $a_{t-1} = a^i$ to index into a matrix of memory vectors to obtain the latest memory $m_i$ associated with $a^i$. This memory vector is passed into the RNN with $f_t$ recurrently. In this way, the state-change resulting from action $a^i$ only influences the embedding $e_i$. Note that we use the same RNN to summarize the state-changes for all actions. Moreover, since $a_{t-1}$ is only used to select the memory, the RNN is agnostic to the action order and focuses only on state-change modeling. Thus, the action embedding produced by the RNN does not contain action semantics, but does carry state-changes information (*i.e.*, action impact).

One technical caveat: the PointNav and ObjectNav tasks both have a special `End` action that denotes that the agent is finished and immediately ends the episode. Unlike the other actions, it makes little sense to apply action drift to `End` as it is completely independent from the agent's actuators. We, in fact, do want our agent to overfit to the semantics of `End`. To this end we employ an Action-Order Encoder which assigns a unique action embedding to the `End` in lieu of the recurrent action embedding. Note that these types of unique action embeddings are frequently used in traditional models to encode action semantics. Finally, we register the recurrent embedding into the action

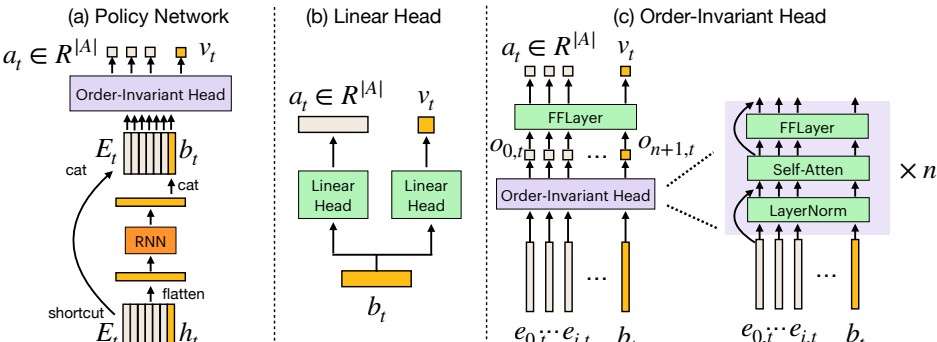

Figure 4: **(a) Policy Network with Order-Invariant Head** first flattens the input and uses an RNN to produce a belief $b$. An Order-Invariant head further processes the action embeddings $E$ and belief $b$ to predict action probability and value. **(b) Linear actor-critic** takes the belief $b$ from RNN to predict action probability and value. **(c) Order-Invariant Head** is invariant to the order of its inputs so the policy predicts the action probability and value based on state-changes (*i.e.*, action impact) instead of action semantics (*i.e.*, a consistent action order).

embedding $e_{i,t}$ via the previous action $a_{t-1} = a^i$, if this action $a^i$ is not the End (*Encode* panel of Fig. 3); otherwise, we register the action embedding $e_{i,t}$ as the output of the Action-Order Encoder.

## 4.3 POLICY NETWORK WITH AN ORDER-INVARIANT HEAD

Standard policy networks in embodied AI use an RNN to summarize state representations $h_t$ through time and an actor-critic head to produce the agent's policy (a probability distribution over actions) and an estimate of the current state's value. Frequently, this actor-critic head is a simple linear function that maps the agent's current beliefs $b_t$ (*e.g.*, the output of the RNN) to generate the above outputs, see Fig. 4 (b). As the matrix multiplications used by this linear function require that their inputs and outputs be explicitly ordered, this imposes an explicit, consistent, ordering on the agent's actions. More specifically, let the weight matrix in the linear mapping be $W = [\mathbf{w}_0| \ldots |\mathbf{w}_n]^T$, then $\mathbf{w}_i$ are the unique set of weights corresponding to the $i$th action $a^i$. Because the $\mathbf{w}_i^T$ is specifically learned for the $i$th action during the training stage, it encodes the semantics of action $a^i$ and thereby prevents the policy from generalizing to unexpected drifts applied to $a^i$. For example, if the action $a^i$ is Rotate(30°), and the policy is learned with training drifts in the range between $[-15°, 15°]$, then policy will fail catastrophically when experiencing an unexpected drift 20° because the policy has overfit to the semantics that the $a^i$ should only be associated with $30° \pm 15°$ rotation.

To address this issue, we propose to equip our policy network with a transformer-based Order-Invariant (OI) actor-critic head, as shown in Fig. 4 (a). The OI head, illustrated in Fig. 4 (c), takes both the belief $b_t$ and the action embeddings $e_{0,t}, \ldots, e_{n,t}$ as input to produce $o_{0,t}, \ldots, o_{n,t}, o_{n+1,t}$, where the $o_{n+1,t}$ is produced by $b_t$. A Feed-Forward Layer then maps the $o_{0,t}, \ldots, o_{n,t}$ to action logits and the $o_{n+1,t}$ to an estimate of the current state's value. Finally, we apply Softmax on the action logits to generate action probabilities. We implement the OI head using a Transformer (Vaswani et al., 2017) encoder without a positional encoding. By design, the extensive weight sharing in a transformer (*e.g.*, the query, key, and value embedding, as well as the Feed-Forward Layer are shared across all the input tokens) means that there are no unique weights assigned to any particular action.

In conclusion, as (1) the transformer uses the same weights for each input embedding, (2) the removal of positional encoding prevents the model from focusing on action semantics, and (3) the set of action embeddings $E$ is used to represent the impact of actions, the proposed AAP can choose the action most useful for its goal despite unexpected drifts. In Sec. 5.1 we show the importance of weight sharing and the action embeddings $E$ by comparing our overall model with two ablations: (i) AAP but with a linear actor-critic head and, (ii) AAP but with no Action-Impact Encoder.

## 4.4 TRAINING STRATEGY

To train our agent, we use DD-PPO (Wijmans et al., 2020) to perform policy optimization via reinforcement learning. To endow the action embeddings $E$ with an ability to describe the state-change between two consecutive observations, we utilize an auxiliary model-based forward prediction loss to optimize the action-impact encoder. In particular, we apply a Feed-Forward Layer (a shared MLP applied to each input independently), which operates on each embedding $\{e_0, \ldots, e_n\}$ independently to predict the agent state-change from the current step $t$ to the next step $t + 1$. As

long as the RNN in the action-impact encoder has been exposed to state-changes resulting from an action, the action-impact encoder can learn to predict the precise agent state change $\Delta s_{t+1}$. Our forward prediction loss is optimized jointly alongside DD-PPO using the same collection of on-policy trajectories collected in standard DD-PPO training. Namely, if $M$ consecutive agent steps are collected during an on-policy rollout, then we apply a simple *MSE* Loss *w.r.t* the ground-truth agent state-changes $\Delta \mathbf{s}^* = \{\Delta s^*_{t-M}, \Delta s^*_{t-M+1}, ..., \Delta s^*_t\}$: $\mathcal{L}_{\text{forward}} = MSE(\Delta \mathbf{s}, \Delta \mathbf{s}^*)$, where $\Delta \mathbf{s} = \{\Delta s_{t-M}, \Delta s_{t-M+1}, ..., \Delta s_t\}$ are the predicted agent state-changes. Therefore, our overall learning objective is $\mathcal{L} = \mathcal{L}_{\text{PPO}} + \alpha \mathcal{L}_{\text{forward}}$, where $\alpha$ controls the relative importance of $\mathcal{L}_{\text{forward}}$. We set $\alpha = 1.0$ in this work. For model optimization details, see Fig. 8 in the Sec. B.

## 5 EXPERIMENTS

In our experiments, we aim to answer the following questions: (1) How does the proposed AAP perform when exposed to expected or modest drifts and how does this performance compare to when it is exposed to unseen or severe drifts? (2) Can the AAP handle extreme cases where some actions are disabled and have no effect on the environment? (3) When faced with multiple disabled actions, can the AAP recognize these actions and avoid using them? (4) Is it important to use the action-impact encoder and OI head jointly or can using one of these modules bring most of the benefit? (5) Qualitatively, how does the AAP adapt to, previously unseen, action drifts?

**Implementation details.** We consider two visual navigation tasks, *PointNav* and *ObjectNav*, using the RoboTHOR framework (Deitke et al., 2020). RoboTHOR contains 75 scenes replicating counterparts in the real-world and allows for rich robotic physical simulation (*e.g.*, actuator noise and collisions). The goals of the two tasks are for the agent to navigate to a given target; in *PointNav* this target is a GPS coordinate and in *ObjectNav* this target is an object category. In the environment, we consider 16 actions in action space $\mathcal{A} = \{\texttt{Move(d)}, \texttt{Rotate}(\theta), \texttt{End}\}$, where $\texttt{d} \in \{0.05, 0.15, 0.25\}$ in meters and $\theta \in \{0°, \pm 30°, \pm 60°, \pm 90°, \pm 120°, \pm 150°, 180°\}$. We set $p = 0.05$m so that we sample movement drifts $d_m \sim U(-0.05, 0.05)$ and $q = 15°$ so that we sample rotation drifts $d_r \sim U(-15°, 15°)$ during training. We then evaluate using drifts $d_m^* \in \{\pm 0.05, \pm 0.1, 0.2, 0.4\}$ and $d_r^* \in \{\pm 15°, \pm 30°, \pm 45°, \pm 90°, \pm 135°, 180°\}$. Note that only $d_m = \pm 0.05$ and $d_r = \pm 15°$ are drifts seen during training. The agent is considered to have successfully completed an episode if it takes the $\texttt{End}$ action, which always immediately ends an episode, and its location is within in $0.2$ meters of the target for *PointNav* or if the target object is visible and within $1.0$ meters for *ObjectNav*. Otherwise, the episode is considered a failure. We use the AllenAct (Weihs et al., 2020) framework to conduct all the experiments. During training, we employ the default reward shaping defined in AllenAct: $R_{\text{penalty}} + R_{\text{success}} + R_{\text{distance}}$, where $R_{\text{penalty}} = -0.01$, $R_{\text{success}} = 10$, and $R_{\text{distance}}$ denotes the change of distances from target between two consecutive steps. See Sec. B for training and model details.

**Baselines.** Each baseline method uses the same visual encoder, goal embedder, RNN and linear actor-critic in the policy network unless stated otherwise. We consider following baselines.
- *EmbCLIP* (Khandelwal et al., 2022) is a baseline model implemented in AllenAct that uses a CLIP-pretrained ResNet-50 visual backbone. It simply uses a state-encoder to obtain hidden state $h_t$, applies a RNN to obtain recurrent belief $b_t$ over $h_t$, and uses a linear actor-critic head to predict an action $a_t$ via $b_t$. This architecture is used by the current SoTA agent for RoboTHOR ObjectNav without action drift (Deitke et al., 2022).
- *Meta-RL* (Wortsman et al., 2019) is an RL approach based on Meta-Learning (Finn et al., 2017). However, since we do not provide reward signal during inference, we cannot simply apply an arbitrary meta-RL method developed in any prior work. Thus, we follow (Wortsman et al., 2019) to employ the same $\mathcal{L}_{\text{forward}}$ in our AAP as the meta-objective during training and inference. We add an MLP after the belief $b_t$ in the EmbCLIP baseline policy to predict the next agent state-change $\Delta s_{t+1}$ to enable the meta-update phase during both training and inference stages.
- *RMA* (Kumar et al., 2021) is a method assessing environmental properties using collected experiences. Although they focus on locomotion tasks, we adapt their idea into our studied problem. At the first stage in RMA training, we input both the movement drift $d_m$ and rotation drift $d_r$ into the model to learn a latent code $c \in \mathcal{R}^8$ of the environment. The policy network then takes $c$ as an additional representation to make a prediction. During the second training stage, we follow (Kumar et al., 2021) to learn a 1-D Conv-Net that processes the past 16 agent states to predict $c$. In this stage, all modules are frozen, except for the 1-D Conv-Net.
- *Model-Based* (Zeng et al., 2021) is a model-based forward prediction approach for the agent state-changes $\Delta \mathbf{s}$. However, they focus on predicting the future agent state-changes associated with differ-

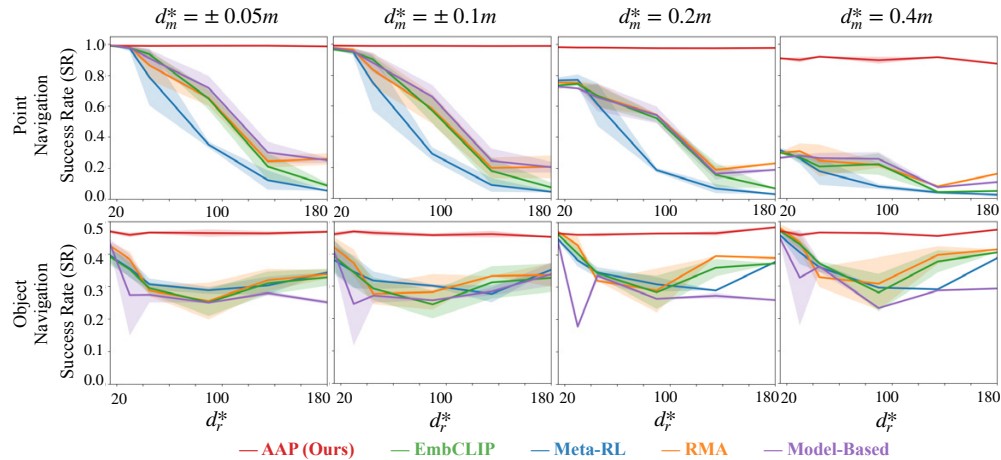

Figure 5: **AAP results.** Top: *PointNav* evaluation. Bottom: *ObjectNav* evaluation. We compare the proposed AAP with baselines, including EmbCLIP, Meta-RL, RMA, and Model-Based. We measure the *SR* over different drifts, including $d_m^* = \{\pm 0.05, \pm 0.1, 0.2, 0.4\}$ and $d_r^* = \{\pm 15°, \pm 30°, \pm 45°, \pm 90°, \pm 135°, 180°\}$. See Sec. A for an example of how our AAP learns to handle unseen drifts by using the Action-Impact Encoder and OI head.

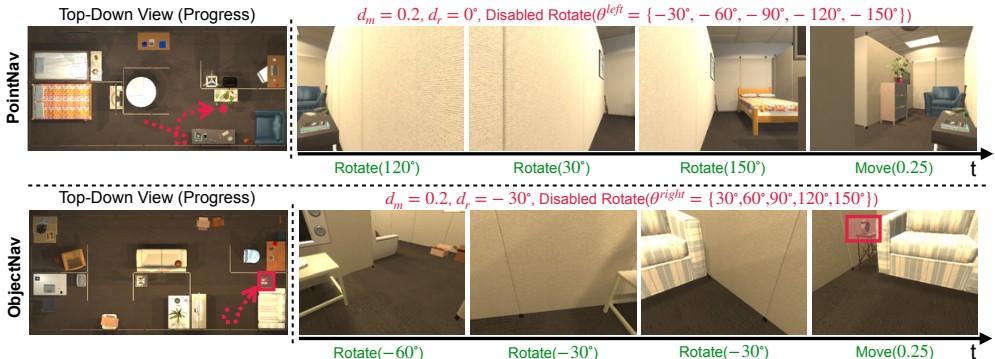

Figure 6: **Qualitative Results.** Examples of *PointNav* (top) and *ObjectNav* (bottom), where $d_m = 0.2$ and $d_r = 0°$ and $d_r = -30°$. Rotate left and rotate right actions are disabled, respectively. The agent adapts by rotating in the other direction to compensate for the disabled actions.

ent actions, and further embed them into action embeddings. Moreover, they use a linear actor-critic to make a prediction which does not fully utilize the advantage of our action-centric approach.

## 5.1 RESULTS

**Evaluation Metrics.** We evaluate all models by their *Success Rate (SR)*; we also report the popular *SPL* (Anderson et al., 2018), *Episode Length*, *Reward*, *Distance to Target*, and *soft-SPL* (Datta et al., 2021) metric in Sec. D. *SR* is the proportion of successful episodes over the validation set. In addition, we report the *Avoid Disabled Actions Rate (ADR)* and *Disabled Action Usage (DAU)* metrics in the **Disabled Actions** experiments. The *ADR* equals the proportion of episodes in which the agent uses no disabled action more than once, and the *DAU* computes the total number of times disabled actions are used averaged across all episodes.

**Different Drifts.** The quantitative results for *PointNav* and *ObjectNav* are shown in Fig. 5. As shown, the *SR* of competing baselines decreases remarkably as the movement drifts $d_m$ and rotation drifts $d_r$ increase from modest ($d_m = \pm 0.05$, $d_r = \pm 15°$) to more severe ($d_m = 0.4$, $d_r = 180°$). Alternatively, our AAP performs consistently across all the drift changes and outperforms baselines significantly. Note that the baselines perform well (*SR*≈100% for *PointNav* and *SR*≈46% for *ObjectNav*) with seen drifts ($d_m = 0.05$, $d_r = 15°$), but our AAP achieves consistent *SR* across all drifts because the action embeddings effectively capture the state-changes associated with actions so the OI head can thereby recognize that, for instance, action $a_i = \texttt{Rotate}(30°)$ becomes another action $a_j = \texttt{Rotate}(120°)$ with rotation drift $d_r = 90°$ during inference. As the magnitude of movement actions are not bounded (unlike rotation actions $\texttt{Rotate}(\theta)$, where $\theta \in [-180°, 180°]$), AAP's performance begins to decrease for very large movement drifts (*e.g.*, $d_m = 0.4$).

Table 1: **Disabled Actions** results in *PointNav* (a,c) and *ObjectNav* (b,d). ↑ and ↓ indicate if larger or smaller numbers are preferred. The experiments are repeated three times.



(a) *PointNav* Success Rate (*SR*)

| SR↑ | disable $\theta^{\text{left}}$ | disable $\theta^{\text{right}}$ |
|---|---|---|
| AAP (Ours) | **96.0±1.1** | **98.0±0.9** |
| EmbCLIP | 11.8±1.4 | 13.0±1.3 |
| Meta-RL | 29.2±4.8 | 28.6±3.4 |
| RMA | 12.5±1.6 | 12.8±4.4 |
| Model-Based | 21.5±10.7 | 14.0±2.8 |

(b) *ObjectNav* Success Rate (*SR*)

| SR↑ | disable $\theta^{\text{left}}$ | disable $\theta^{\text{right}}$ |
|---|---|---|
| AAP (Ours) | **31.2±3.3** | **38.8±3.0** |
| EmbCLIP | 12.8±3.2 | 23.8±1.7 |
| Meta-RL | 18.7±1.7 | 26.2±1.9 |
| RMA | 9.3±6.8 | 24.2±2.1 |
| Model-Based | 19.2±1.2 | 9.9±1.2 |

(c) Analysis of disabled actions in *PointNav*

| ADR↑ / DAU↓ | disable $\theta^{\text{left}}$ | disable $\theta^{\text{right}}$ |
|---|---|---|
| AAP (Ours) | **12.5 / 27.8** | **18.4 / 15.7** |
| EmbCLIP | 0.3 / 93.9 | 1.0 / 93.3 |
| Meta-RL | 0.7 / 83.1 | 0.6 / 85.2 |
| RMA | 0.4 / 94.4 | 0.5 / 95.2 |
| Model-Based | 0.7 / 89.4 | 0.6 / 96.1 |

(d) Analysis of disabled actions in *ObjectNav*

| ADR↑ / DAU↓ | disable $\theta^{\text{left}}$ | disable $\theta^{\text{right}}$ |
|---|---|---|
| AAP (Ours) | **3.2 / 41.4** | **5.3 / 28.5** |
| EmbCLIP | 0.6 / 90.5 | 2.9 / 71.2 |
| Meta-RL | 2.0 / 80.1 | 2.2 / 84.5 |
| RMA | 0.6 / 95.9 | 2.6 / 73.3 |
| Model-Based | 2.5 / 81.4 | 1.5 / 89.8 |



**Disabled Actions.** We consider two experimental settings to evaluate how models perform when some actions are disabled (*e.g.*, due to to a malfunctioning motors or damaged wheel). In the first setting we disable the 5 rotation angles $\theta^{\text{right}} \in \{30°, 60°, 90°, 120°, 150°\}$ so that the agent cannot rotate clockwise and, in the second setting, we disable $\theta^{\text{left}} \in \{-30°, -60°, -90°, -120°, -150°\}$ so that the agent can only rotate clockwise. We otherwise set $d_m = 0.2$m and $d_r = 0°$ in this experiment. The *SR* for *PointNav* and *ObjectNav* are shown in Tab. 1a and Tab. 1b. As shown, there is a huge difference between the baselines and AAP; for instance, for the *PointNav* task, AAP achieves near 100% *SR* while the best competing baseline (Meta-RL) achieves <30%. In addition, AAP outperforms the baselines by at least 12.5% on average in *ObjectNav*. We further report the *ADR* and *DAU* metrics for this task in Tab. 1c and Tab. 1d; from these tables we see that AAP is very effective at recognizing what actions are disabled through the proposed action embeddings $E$, see *ADR*, and efficiently avoids using such actions, see *DAU*. Fig. 6 shows an example with $\theta^{left}$ on *PointNav* and another one with $\theta^{right}$ on *ObjectNav*. See more examples in Sec. E and Fig. 14.

**Ablation Studies.** We investigated which module in the AAP contributes the most to the performance, or if they are all necessary and complementary to one another. We conducted the ablation studies on *Point-Nav* by comparing our proposed AAP to: (1) "*LAC*", a variant of AAP that uses a linear actor-critic head instead of our OI head and (2) "*Action-Semantics*", a variant of AAP that uses the OI head without the Action-Impact Encoder (so there are unique learnable embeddings for each unique action). The results are shown in Fig. 7. Although *Action-Semantics* achieves similar *SR* to AAP when $d_m = \pm0.1m$ and $d_r \in \{\pm15°, \pm30°\}$, its performance starts decreasing as the drifts become more dramatic. In contrast, the poor *SR* of *LAC* confirms our OI head is far better leverage the predicted action embeddings than a linear actor-critic.

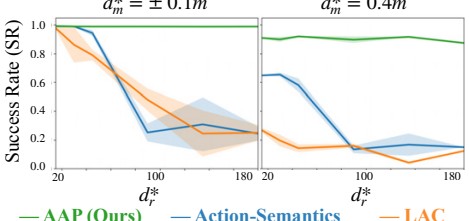

Figure 7: **Ablation studies.** We conducted the ablation studies on *PointNav* by comparing AAP to: (1) "*LAC*", a variant of AAP that uses a linear actor-critic head instead of our OI head and (2) "*Action-Semantics*", a variant of AAP that uses the OI head without the Action-Impact Encoder.

**Real-World Experiments.** We train our AAP and EmbCLIP on ProcTHOR 10k (Deitke et al., 2022) for *ObjectNav* with training drifts $d_m$ and $d_r$, then evaluate in a real-world scene from RoboTHOR (Deitke et al., 2020). In an exploratory experiment, the goal is to find `Apple` and `Chair` from 2 different starting locations (20 episodes) at inference. We inject

Table 2: **Real-World Experiments** results in RoboTHOR on *ObjectNav*.

| Model | SR↑ |
|---|---|
| AAP (Ours) | 65% |
| EmbCLIP trained *w.* drifts | 30% |
| EmbCLIP | 25% |

the drifts ($d_m^* \in \{-0.05m, 0m, 0.1m\}$ and $d_r^* \in \{0°, \pm60°\}$) by adding the drifts to the low-level commands used by the robot. Tab. 2 shows AAP performs more robustly against these unseen drifts.

# 6 CONCLUSION

We propose Action Adaptive Policy (AAP) to adapt to unseen effects of actions during inference. Our central insight is to embed the impact of the actions on-the-fly rather than relying on their training-time semantics. Our experiments show that AAP is quite effective at adapting to unseen action outcomes at test time, outperforming a set of strong baselines by a significant margin, and can even maintain its performance when some of the actions are completely disabled.

ACKNOWLEDGMENTS

We thank members of the RAIVN lab at the University of Washington and the PRIOR team at the Allen Institute for AI for their valuable feedback on early versions of this project. This work is in part supported by NSF IIS 1652052, IIS 17303166, DARPA N66001-19-2-4031, DARPA W911NF-15-1-0543, J.P. Morgan PhD Fellowship, and gifts from Allen Institute for AI.

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

SUPPLEMENTARY MATERIAL

## A    GENERALIZATION OF AAP TO UNSEEN DRIFTS

We provide an example here to illustrate how our AAP generalizes to unseen drifts during the testing stage. In the training stage, the rotation actions cover all degrees $\theta \in \{-150° \pm 15°, -120° \pm 15°, ..., 150° \pm 15°, 180° \pm 15°\}$ despite with small drifts $d_r = \pm 15°$, our AAP observed all possible rotation **outcomes** over $[-180°, 180°]$. Therefore, during the testing stage, taking $d_r^* = 180°$ as an example, our policy can recognize the effect of $a^i = \text{rotate}(30° + 180° \text{drifts})$ is equivalent to the effect of another action $a^j$: $\text{rotate}(-150°)$ which was seen in the training stage. Likewise, since the movement magnitude $d \in \{0.05 \pm 0.05, 0.15 \pm 0.05, 0.25 \pm 0.05\}$ can cover $[0, 0.3]$, our model performs well when the movement magnitude is within this range. This example highlights the core technical contributions, namely the action-impact embedding and OI head, in this work. With the action-impact embedding and OI head, we prevent the policy from **remembering** the action semantics, and allow it to focus on modeling the effect of actions.

## B    TRAINING PIPELINE, HYPERPARAMETERS, AND TIME COMPLEXITY

The training pipeline for forward pass and backward update is shown in the Fig. 8. During training, we use the Adam optimizer and an initial learning rate of $3e-4$ that linearly decays to 0 over 75M and 300M steps for the two tasks, respectively. We set the standard RL reward discounting parameter $\gamma$ to 0.99, $\lambda_{gae}$ to 0.95, and number of update steps to 128 for $\mathcal{L}_{\text{PPO}}$. The $\alpha$ for $\mathcal{L}_{\text{forward}}$ is set to 1. Finally, we train the policy for 75M and 200M steps for *PointNav* and *ObjectNav* respectively and evaluate the model every 5 million steps.2

- *Time complexity*. Theoretically, the time-complexity for the OI transformer head is $O(n^2)$, where the $n$ is the number of actions in the action space. However, as the number of actions is usually fairly small, our AAP doe snot suffer from the same problems frequently plague transformer-based models with their $n^2$ complexity in the sequence length. At runtime, we evaluate models by a personal desktop with a Intel i9-9900K CPU, 64G DDR4-3200 RAM, 2 Nvidia RTX 2080 Ti GPUs. The framework (w/ our AAP) spends $\approx 35$ minutes evaluating 1.8k val episodes with 5 parallel processes on the Point Navigation task. The average episode length is $\approx 117$. As a result, the FPS (or interaction per second) is $\approx 100.3$. During the training phase, we used an AWS machine with 48 vCPUs, $187G$ RAM, and 4 Nvidia Tesla T4 GPUs to train the policy. The FPS (or interaction per second) is $\approx 400$ for our AAP.

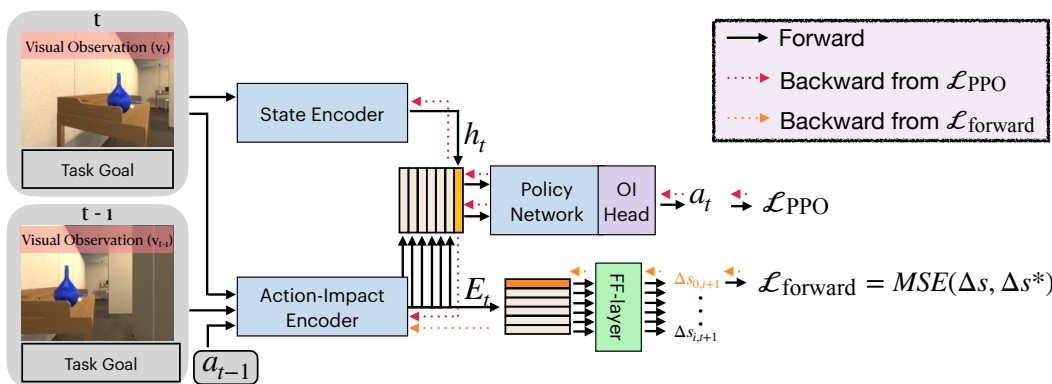

Figure 8: **Training Pipeline.** The forward pass is in black color, the backward from $\mathcal{L}_{\text{PPO}}$ is in red color, and the backward from $\mathcal{L}_{\text{forward}}$ is in orange color.

## C    MODEL ARCHITECTURE DETAILS

- *Ours*. We follow (Khandelwal et al., 2022) and set the dimension of the belief $b_t$ and hidden dimension to 512, and the output of ResNet is a 1568-dim goal-conditioned visual embedding. In

the Action-Impact Encoder, the dimension of r* is 1024-dim for the Point Navigation and 3136 for the Object Navigation. The goal embedding is 8-dim, the hidden for the GRU is 128-dim, and the action embedding is also 128-dim. In the policy network, the GRU is 512-dim. The OI-Head consists of 6 layers of Transformer Encoder with 16 heads, 512-dim hidden, and 512-dim output.

- *EmbCLIP* (Khandelwal et al., 2022). The model architecture is the same one used in (Khandelwal et al., 2022) which is open-sourced on AllenAct (Weihs et al., 2020)[2].

- *Meta-RL* (Wortsman et al., 2019). The model architecture is the same as the one used in (Khandelwal et al., 2022), except that we add a MLP after the belief b to predict the agent-state change for meta-learning. Instead of performing meta-update every 6-steps, we perform meta-update every 20-steps, because the maximum allowed number of steps per episode was increased from 200 to 500 in our navigation tasks. The meta-update learning rate was set to $10^{-4}$ (same as (Wortsman et al., 2019)).

- *RMA* (Kumar et al., 2021). The main model architecture is the same as the one used in (Khandelwal et al., 2022), except that we add a module to learn latent code and an adaptation module to perform online adaptation. More specifically, we follow (Kumar et al., 2021) to encode the latent code z (8-dim) by a 3 layer MLP ($2 \rightarrow 256 \rightarrow 128 \rightarrow 8$) from motion and rotation drifts in the first training stage. In the adaptation stage, we follow (Kumar et al., 2021) to learn the adaptation module to predict latent code from 16 past agent-states. In details, the adaptation module consists of an MLP embedding ($3 \rightarrow 14 \rightarrow 32 \rightarrow 32$) and a 3-layer 1D Conv ($[32, 32, 8, 2] \rightarrow [32, 32, 3, 1] \rightarrow [32, 8, 3, 1]$) over 16 past states.

- *Model-Based* (Zeng et al., 2021). The model architecture is the same as our proposed model, except that the model produces action embeddings from the predicted state changes. In this way, this baseline can utilize the next state predictions associated with different actions for planning. In addition, it is important to note that the Policy Network for this baseline is the linear actor-critic, instead of the proposed OI-Head.

## D  MORE QUANTITATIVE RESULTS

We show more quantitative results for *PointNav* and *ObjectNav* in Fig. 9, Fig. 10, Fig. 11, Fig. 12, and Fig 13 by *SPL*, *Episode Length*, *Reward*, *Distance to Target*, and *soft-SPL*As shown, the *SPL*, *soft-SPL*, and *Episode Length* of competing baselines achieve better results than AAP at the modest drifts ($d_r = \pm 15°, \pm 30°$), but they perform significantly worse at more severe drifts. Note that our AAP initially has no idea about the state-changes resulting from each action, it has to spend more time exploring the effects of actions in the beginning of a new episode. As a result, it results in a longer episode length comparing to the baselines with seen drifts. On the other hand, our AAP achieves consistent *SPL*, *Episode Length*, *Reward*, *Distance to Target*, and *soft-SPL* across all the drift changes and outperforms baselines remarkably. It is because AAP does not memorize action semantics, but relies on its experiences in the inference time to embed the action embeddings $E$ on-the-fly. To collect useful experiences, the agent has to explore the environment and update the embeddings accordingly. Thereby, our AAP takes more time to understand the state-changes, while the baselines can achieve better results based on its learned action semantics in the scenarios with the modest drifts.

- *Comparison to (Adapted) Model-Based Meta-RL (Nagabandi et al., 2018)*. Unfortunately, a direct comparison to (Nagabandi et al., 2018) is not feasible as, (a) we focus on point navigation and object navigation in a clustered scene, but (Nagabandi et al., 2018) focuses on locomotion control for a simple straight or curve line movement, (b) our main observation is visual observation, but (Nagabandi et al., 2018) uses agent's state, and (c) we don't use model-predictive controller to rollout H time horizon future to make a decision, because the agent cannot access rewards or distance to the goal during the testing time, so it is not straightforward to implement a simple objective function in MPC for point navigation or object navigation task (we would argue that doing so is a research topic in itself).

---

[2]We use the open-sourced code to train EmbCLIP `https://github.com/allenai/allenact/blob/main/projects/objectnav_baselines/experiments/robothor/clip/objectnav_robothor_rgb_clipresnet50gru_ddppo.py`.

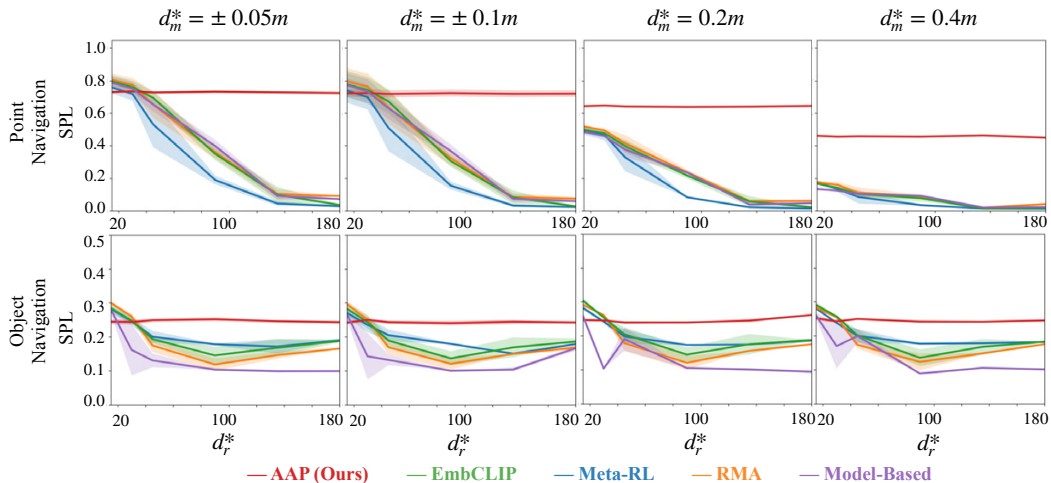

Figure 9: **Quantitative result using the SPL metric.** The *SPL* (Anderson et al., 2018) is defined as $\frac{1}{N}\sum_{n=1}^{N}S_n\frac{L_n}{max(P_n,L_n)}$, where $N$ is the number of episodes, $S_n$ denotes a binary indicator of success in the episode $n$, $P_n$ is the path length, and $L_n$ is the shortest path distance in episode $n$. Top: *PointNav* evaluation. Bottom: *ObjectNav* evaluation. We compare the proposed AAP with baselines, including EmbCLIP, Meta-RL, RMA, and Model-Based. We measure the *SPL* over different drifts, including $d_m^* = \{\pm 0.05, \pm 0.1, 0.2, 0.4\}$ and $d_r^* = \{\pm 15°, \pm 30°, \pm 45°, \pm 90°, \pm 135°, 180°\}$.

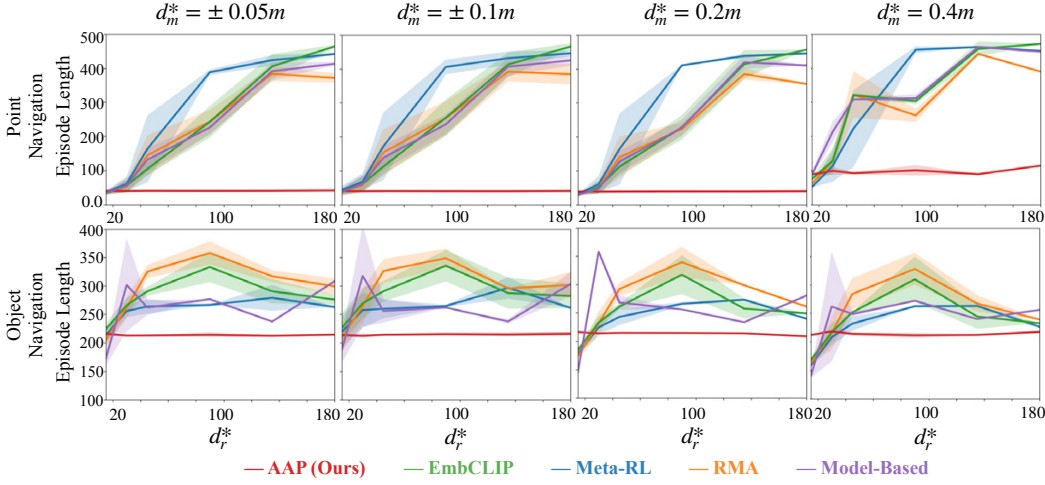

Figure 10: **Quantitative result using the Episode Length metric.** Top: *PointNav* evaluation. Bottom: *ObjectNav* evaluation.

However, to have a closer implementation based on the idea proposed in (Nagabandi et al., 2018), we combine our model-based baseline with the meta-RL baseline by employing the agent state prediction to perform meta-learning on the model-based module. The results for point navigation are shown in Fig. 13 in the green color by *soft-SPL*. Although this (Adapted) Model-Based Meta-RL outperforms EmbCLIP slightly facing larger drifts, it is still not able to overcome the severe drifts. It, again, highlights the effectiveness of our AAP with two major technical contributions, the action-impact embedding and the OI head, proposed in this work.

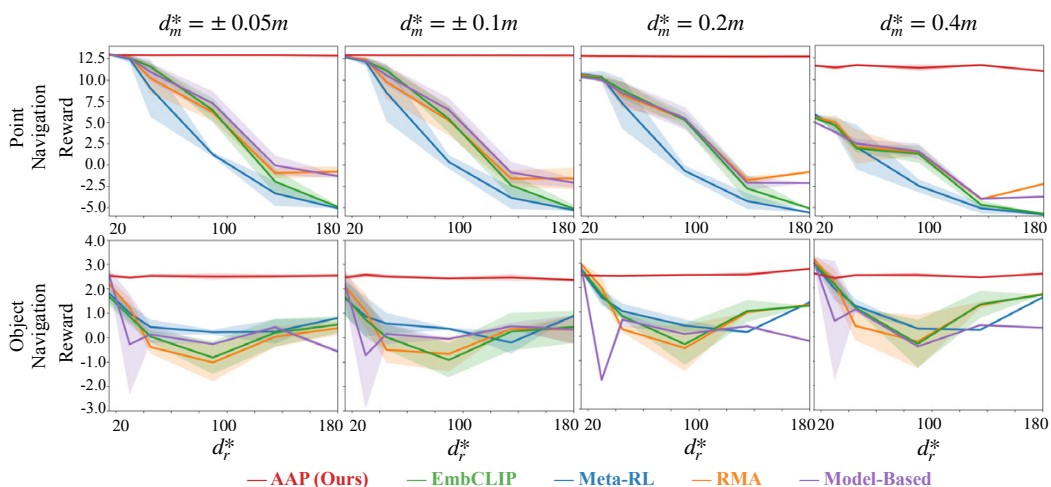

Figure 11: **Quantitative result using the Reward metric.** Top: *PointNav* evaluation. Bottom: *ObjectNav* evaluation.

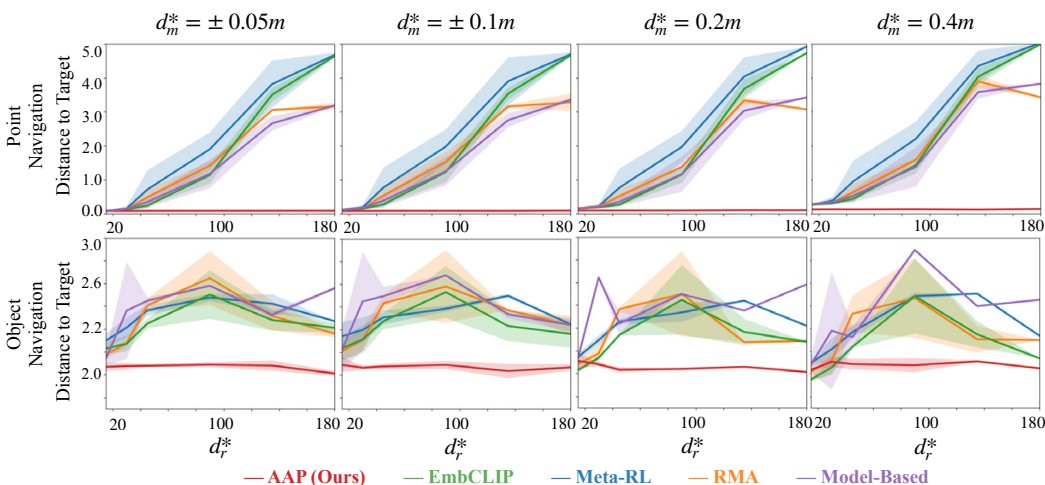

Figure 12: **Quantitative result using the Distance to Target metric.** Top: *PointNav* evaluation. Bottom: *ObjectNav* evaluation.

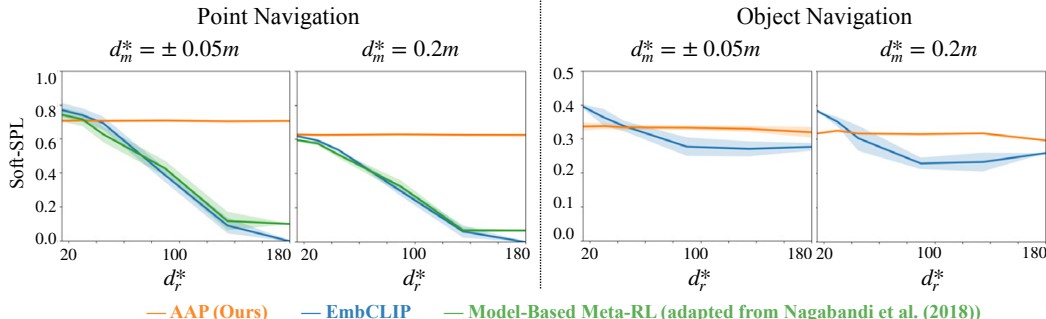

Figure 13: **Quantitative result using the soft-SPL.** *soft-SPL* (Datta et al., 2021) is defined as $\frac{1}{N}\sum_{n=1}^{N}(1 - \frac{d_{n,\text{termination}}}{d_{n,\text{start}}})\frac{L_n}{max(P_n, L_n)}$, where $N$ is the number of episodes, $d_{n,\text{termination}}$ and $d_{n,\text{start}}$ denote the (geodesic) distances to target upon termination and start in the episode $n$, $P_n$ is the path length, and $L_n$ is the shortest path distance in episode $n$. Left: *PointNav* evaluation. Right: *ObjectNav* evaluation.

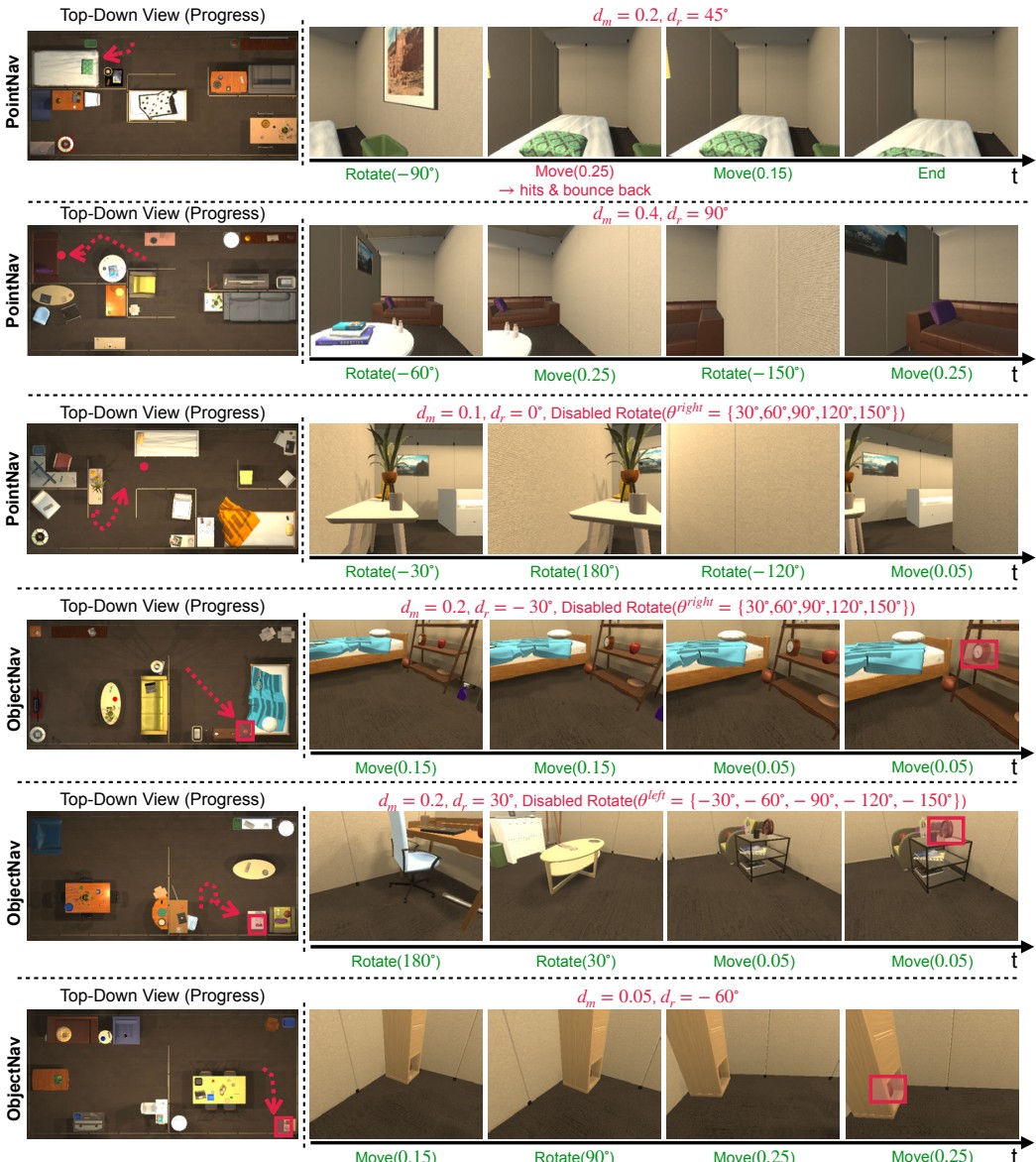

Figure 14: **Qualitative Results.** Examples of *PointNav* (top tree rows) and *ObjectNav* (bottom three rows). The agent adapts by rotating in the other direction to compensate for the disabled actions. The first example shows the agent adopts a smaller movement magnitude to avoid the collision. The second example shows smooth moves by a right-turn, a move, another left-turn, and a final move to dodge the white table. The third example shows the agent uses three left turns with different angles to make a disabled right turn. The fourth example shows the agent slows its movement magnitude as it approaches the target object in the clustered area. The fifth example shows the agent uses two left-turn with different angles to make a disabled right turn to find the target object. The final example shows the agent uses a larger movement magnitude to move toward the target object in a relatively open area.

## E QUALITATIVE RESULTS

We show more qualitative results in *PointNav* and *ObjectNav* in Fig. 14. The first example shows the agent adopts a smaller movement magnitude to avoid the collision. The second example shows smooth moves by a right-turn, a move, another left-turn, and a final move to dodge the white table. The third example shows the agent uses three left turns with different angles to make a disabled right turn. The fourth example shows the agent slows its movement magnitude as it approaches the target

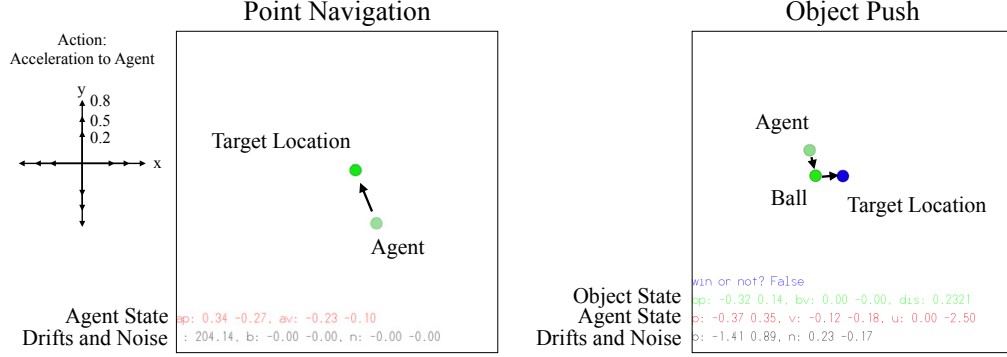

Figure 15: **MPE environment in PettingZoo**. We modify the MPE environment to simulate *Point-Nav* and *ObjectPush* tasks. The goal for *PointNav* is to move the agent to the target location and the goal for *ObjectPush* is to move the agent to push the ball to the target location. The action space consists of 3 different accelerations towards 4 directions.

object in the clustered area. The fifth example shows the agent uses two left-turn with different angles to make a disabled right turn to find the target object. The final example shows the agent uses a larger movement magnitude to move toward the target object in a relatively open area.

## F  RESULTS ON THE MODIFIED PETTINGZOO ENVIRONMENT

We modify the PettingZoo (Terry et al., 2020) to verify the effectiveness of our AAP in a different environment with state observations only. We modify the MPE environment to simulate *PointNav* and *ObjectPush* tasks, shown in Fig. 15. The MPE environment provides simple simulation of collision force when objects are too close to each other. The goal for *PointNav* is to move the agent to the target location by applying acceleration to the agent. The goal for *ObjectPush* is to move the agent to push the ball to the target location by applying acceleration to the agent. The agent has to collide to the ball to perform the *push*. As shown in the figure, there are 12 actions in action space, including Accelerate ($\text{mag}_x$, $\text{mag}_y$), where ($\text{mag}_x$, $\text{mag}_y$)= $\{[\pm 0.2, 0], [\pm 0.5, 0], [\pm 0.8, 0], [0, \pm 0.2], [0, \pm 0.5], [0, \pm 0.8]\}$, corresponding to 3 different accelerations towards 4 different directions. For *PointNav*, the state space is 6-dim, including agent's position $(p_x, p_y)$, agent's velocity $v_x, v_y$, GPS sensor $(\Delta p_x, \Delta p_y)$; for *ObjectPush*, the state space is 10-dim, including agent's position $(p_x, p_y)$, agent's velocity $(v_x, v_y)$, object's position $(o_x, o_y)$, object's velocity $(o_{v_x}, o_{v_y})$, GPS sensor $(\Delta p_x, \Delta p_y)$. Our drifts setting for training and inference stage are formulized as follows,

**During training.** At the start of a training episode, we sample the rotation drift $d_r \sim U(-90°, 90°)$, where $U(\cdot, \cdot)$ denotes a uniform distribution.

**During inference.** At the start of an evaluation episode, we evaluate our AAP and baselines with an unseen rotation drift $d_r^* \in \{\pm 120°, \pm 135°, \pm 150°, \pm 180°\}$.

With a rotation drift $d_r$, the environment applies the $d_r$ to the input action by rotating the acceleration direction:

$$\text{mag}_{\text{env}} = \begin{bmatrix} cos(d_r) & sin(d_r) \\ -sin(d_r) & cos(d_r) \end{bmatrix} \text{mag}_{\text{input}}^T. \quad (1)$$

In this case, the actual direction of acceleration would be drifted according to the rotation drift $d_r$.

**Models.** The details about our AAP and the considered baselines are as follows,

- *GRU* simply uses a 4-layer MLP ($|\text{state}| \rightarrow 64 \rightarrow 64 \rightarrow 64]$) as the state-encoder to obtain a 64-dim hidden state $h_t$, applies a RNN (GRU) to obtain a 64-dim belief $b_t$ over $h_t$, and uses a linear actor-critic head to predict an action $a_t$ via $b_t$.

- *LAC* is a variant of our AAP that uses a linear actor-critic head instead of our OI head.

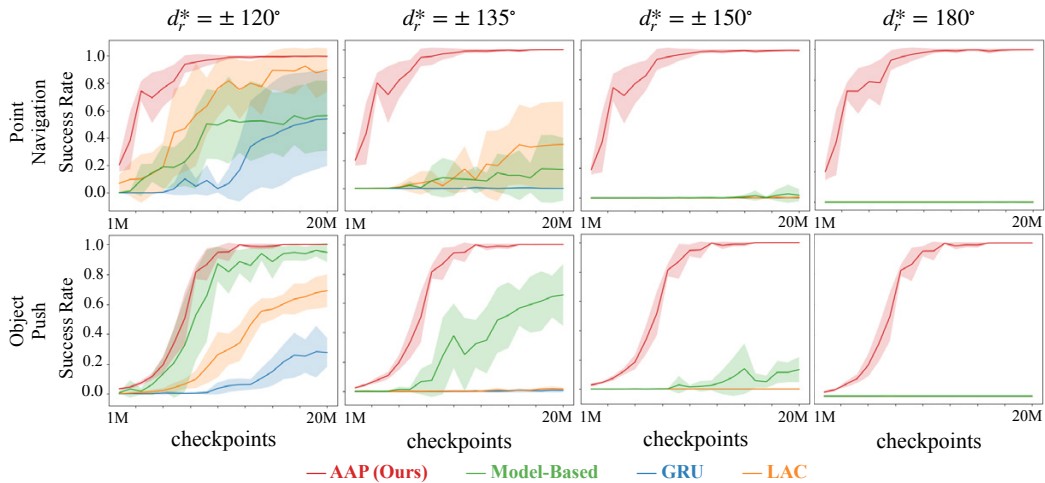

Figure 16: **Success Rate in MPE environment**. Top: *PointNav* evaluation. Bottom: *ObjectPush* evaluation. We train each model by 3 different random seeds and show the average success rate and $1\times$ standard deviation in this figure.

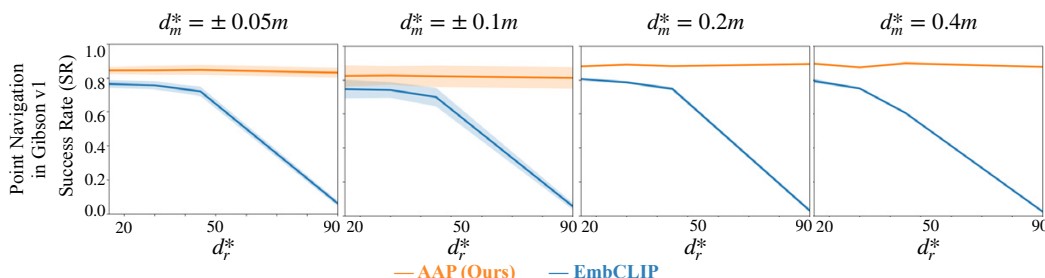

Figure 17: **Success Rate on Point Navigation in Habitat Environment.** We compare the proposed AAP with EmbCLIP. The evaluation drifts are unseen during the training stage, including $d_m^* \in \{\pm0.05m, \pm0.1m, 0.2m, 0.4m\}$ and $d_r^* \in \{\pm15°, \pm30°, \pm45°, \pm90°\}$ The experiments are conducted in Habitat-Lab `v0.2.1` on Point Navigation with Gibson v1.

- *Model-based* is a model-based forward prediction approach for the agent state-changes $\Delta\mathbf{s}$ associated with different actions. It further embeds the prediction into action embeddings. Finally, this baseline uses a linear actor-critic to make a prediction.

- *Our AAP* uses the same 4-layer MLP used in *GRU baseline* to produce state encoding for previous state and the current state, respectively. Then, same as our AAP used in AI2THOR, the action-impact encoder uses a RNN (GRU) to produces the $n \times 64$-dim action impacts embedding $E$ associated with different actions. Later, the model uses a RNN (GRU) in the policy obtain a 64-dim belief $b_t$. Finally the OI head operates on the concatenation of $E$ and $b_t$ to make a prediction.

**Results.** We train every model by PPO with the Adam optimizer and an initial learning rate of $1e-3$ that linearly decays to 0 over 20M. We set the standard RL reward discounting parameter $\gamma$ to 0.99, $\lambda_{gae}$ to 0.95, and number of update steps to 200 for $\mathcal{L}_{PPO}$. The $\alpha$ for $\mathcal{L}_{forward}$ is set to 1. Finally, we train each model by 3 different random seeds and obtain the average success rate and $1\times$ standard deviation. We evaluate models every 1M steps (checkpoint). The evaluation results are presented in Fig. 16. As shown in the figure, our AAP performs consistently well across all rotation drifts on both tasks in the modified MPE environment. The best baselines cannot even achieve any success rate when facing the extremest rotation drift $d_r^* = 180°$. It verifies that the effectiveness of the proposed AAP on a general reinforcement learning environment. We will release the code for this modified environment and experiments as well.

## G  RESULTS ON THE HABITAT ENVIRONMENT

To conduct the experiments in Habitat (Savva et al., 2019), we made a small change in the Habitat's `Move_Forward` action, where every `Move_Forward` only moves the agent by 0.01m. If the action $a^i$ = `Move_Forward(0.25)`, the environment would move the agent by 25 times. In this way, we can implement the movement drifts as we did in AI2-THOR. For the rotation action, we do not use the default `TURN_RIGHT` or `TURN_LEFT` action. We instead directly compute the quaternion to implement the continuous rotation with rotation drifts. In this modified Habitat, we train the EmbCLIP and our AAP on Point Navigation task in Gibson `v1` (Xia et al., 2020). The simulator is Habitat-Lab `v0.2.1`. The training settings are the same as the settings used in AI2-THOR, including the same training drifts, same learning schedule, same optimization algorithm, and learning objective. During the evaluation, we evaluate the model on only movement drift $d_m^* \in \{\pm 0.05m, \pm 0.1m, 0.2m, 0.4m\}$ and rotation drift $d_r^* = \{\pm 15°, \pm 30°, \pm 45°, \pm 90°\}$. As shown in Fig. 17, our AAP performs consistently across all movement and rotation drifts. However, the EmbCLIP is struggling with larger drifts.

