# OpenReview forum: "Moving Forward by Moving Backward: Embedding Action Impact over Action Semantics"
_ICLR.cc/2023/Conference — ICLR 2023 notable top 5%_

### Official Review · Reviewer_AaUJ · 2022-10-20

**Confidence:** 5
**Correctness:** 3
**Technical Novelty And Significance:** 3
**Empirical Novelty And Significance:** 3
**Recommendation:** 8

**Clarity, Quality, Novelty And Reproducibility:**

Novelty: The proposed model is novel. I don't think this is a problem. I have already described in my review the strengths, and the architecture described in the manuscript seems not to have been tried before. In addition, the proposed problem is also novel. If the paper is accepted, there may be more authors exploring the same problem in the AI2THOR environment. For this, the experimental setup must be crystal clear.

Clarity and Quality: The article is well written, and the quality of the figures deserves a special mention.


Reproducibility:
-Code will be released. Moreover, publicly available environments are used (RoboTHOR in the AI2-THOR environment).

**Strength And Weaknesses:**

Regarding the strengths of the paper:

+It is a well-written article. The editing work is excellent. The figures are of great quality, as well as being very informative.

+Section 2 is approached effectively. Previous work is presented and it is explicitly stated why the proposed ideas are different.

+The paper includes a clear formulation of the problem. I think it is right not to consider that the drifts observed during the test have not been observed during training. In this way it is possible to infer what generalization capacity the model has. However, I also see no problem in dealing with a scenario in which training and test share the drifts, if they are chosen randomly in both.

+As for the model itself, I would point out that it uses encoders that are the state of the art (CLIP). The combination of functional blocks proposed in Figure 2 is also novel. In particular, I have not found previous work that combines a model such as the Action-Impact Encoder, with a policy network and the OI invariant head detailed in the paper.

+Finally, it should be noted that the experiments have been carried out in a publicly available environment. Code will be released, and results will be easily replicated. I also appreciate the informative ablation study. From it, it is clear to me the impact of the OI head in the process.

The main weaknesses of the work are as follows:
-Comparison with state of the art models is incomplete. The experimental evaluation offers a comparison with multiple baselines, bot none of them focuses on the problem addressed in the paper. In other words, the baselines have not been designed to tackle the rotation and movement drifts. This would explain their (low) performance for some settings. Possibly only RMA (Kumar et al., 2021) has been slightly adapted to explicitly deal with the proposed problem. It is actually the one reporting the best results for the ObjectNav task (even better than the approach in the paper for low rotation drifts). The meta-learning model of Nagabandi et al. (2018) could have been explored. Note that is is different from the meta-learning baseline proposed. Does the proposed model improves the performance of an state-of-the-art model that using meta-learning is able to optimize the model at the testing phase? This could be an interesting question to answer. This would help sell the paper's contributions. Overall, I'm missing a direct comparison with other models that explicitly consider the same problem.

-The experimental evaluation shows the following inconsistencies:

a) Figure 9 and Figure 5 show different winners. I mean, for example, for the problem of ObjectNav, the approach presented in this paper reports the best SR, but when one changes the evaluation metric to the SPL, then the winners (for low rotation drifts) are the rest of the models. The two metrics are different, I know them quite well. But SPL is quite informative for the problem considered, and I am concerned that the proposed model is not able to approach to the optimal route, when the others do. The other well established soft-SPL could be used. Have the authors tried it?

b) Overall, across all figures, it seems that the model is not affected by the rotation drifts. Even for the 180 degrees drift! The SR should necessarily decrease (slightly) when the agent is exposed to such high drifts. Moreover, to me, it is unclear how the results can be the ones they are, when during training the only seen drifts are +/- 0.05 and +/- 15 for the movement and rotation, respectively. The paper needs the better explain the results that are being reported.

c) Figure 5 again, let's analyze one more time. For the ObjectNav experiment, it seems that the displacement drift does NOT affect the
baselines, if the rotation drift is kept at 20º. Here there is no clear improvement with respect to the rest of the models. This aspect is worrying, as this experiment is more relevant than PointNav. Moreover, the result (of the baselines) is better the more drift, for models that have not been trained for it. These results are confusing, and in my opinion do not allow to offer a clear conclusion.


**Summary Of The Paper:**

This paper addresses the problem of designing a reinforcement learning (RL) policy that lets the agents to adapt to changes in the form of rotation and movement drifts.
The manuscript describes a novel approach which includes an state encoder and an action impact encoder. The former is a standard state encoder where a visual encoder and an embedder for the task are concatenated. For this work state-of-the-art state encoders are used (Deitke et al. (2020), and Savva et al. (2019)).
The action-impact encoder is the new block proposed in this work. It receives two consecutive frames as well as the previous action, and its objective consists in producing a set of action embeddings that summarize the impact of the actions performed by the agent in an episode. Technically, this is done with a combination of a RNN and the state encoder. Finally, for the policy, a transformed-based approach is proposed. It is called the Order-Invariant (OI) head. Thanks to the transformer, this OI head is independent of the sequence of its inputs and analyzes the agent's belief together with the action embeddings to help it decide which action will have the greatest influence on achieving its objective.
Experiments are provided in RoboTHOR framework in the AI2-THOR environment. Two navigation tasks are evaluated: PointNav and ObjectNav. Results show some improvements of the proposed approach compared to a set of baselines when the Success Rate metric is used, especially when the rotation drift appears.

**Summary Of The Review:**

Overall, I think this is an interesting paper. The problem it proposes is relevant, and providing a clear experimental setup for the rest of the community would be of great value. However, there are some aspects that I did not find convincing. The comparison with baselines and other state-of-the-art models is poor. This is usually the case when a problem is totally new, but this would not be the case. I think more work should be done on this aspect, especially for a paper that aspires to be published in the ICLR.
In addition, as I have described, the results present some inconsistencies. I would ask the authors to address these in their feedback, so that a reasoned decision can be made regarding the manuscript.

---

> ### Author Response · Authors · 2022-11-18
> **Response to Reviewer AaUJ**
>
> We thank you for the valuable feedback. In the following, we use W for Weakness.
>
> **W0. The meta-learning model of Nagabandi et al. (2018) could have been explored.**
>
> Unfortunately, a direct comparison to Nagabandi et al. (2018) is not feasible as, (a) we focus on point navigation and object navigation in a cluttered scene, but Nagabandi et al. (2018) focuses on locomotion control for a simple straight or curve line movement, (b) our main observation is visual observation, but Nagabandi et al. (2018) uses agent’s state, and (c) we don’t use model-predictive controller to rollout H time horizon future to make a decision, because the agent cannot access rewards or distance to the goal during the testing time, so it is not straightforward to implement a simple objective function in MPC for point navigation or object navigation task (we would argue that doing so is a research topic in itself). In addition, we want to point out that our Meta-RL baseline is able to optimize the model at the testing phase using the agent state prediction.
>
> However, to have a closer implementation based on the idea proposed in Nagabandi et al. (2018), we combine our model-based baseline with the meta-RL baseline by employing the agent state prediction to perform meta-learning on the model-based module. The results for point navigation are shown in Fig.13 in Appendix D. Although this (Adapted) Model-Based Meta-RL outperforms EmbCLIP slightly facing larger drifts, it is still not able to overcome the severe drifts. It, again, highlights the effectiveness of our AAP with two major technical contributions, the action-impact embedding and the OI head, proposed in this work.
>
> **W1. Figure 9 and Figure 5 show different winners. I mean, for example, for the problem of ObjectNav, the approach presented in this paper reports the best SR, but when one changes the evaluation metric to the SPL, then the winners (for low rotation drifts) are the rest of the models. The two metrics are different, I know them quite well. But SPL is quite informative for the problem considered, and I am concerned that the proposed model is not able to approach to the optimal route, when the others do. The other well established soft-SPL could be used. Have the authors tried it?**
>
> That is correct, our method may not choose the optimal route. This is because our method starts the episode without any explicit knowledge about what its actions do and so it must take some initial actions to allow it to produce meaningful action impact embeddings for these actions. This need to "explore" the action space means that our method will be slightly less efficient than competitors when there are no (or only slight) action drifts. Note that in any realistic, deployed, setting this problem would quickly be amortized away as the agent would almost certainly be asked to do more than one navigation task sequentially and action embeddings can be saved across trials in the same environment. Given the above, the Success Rate metric better shows the strength of our approach. Due to time constraints, we have added the episode length, reward, and distance to target metrics from our evaluation logs to Appendix D (Fig. 10, Fig. 11, and Fig. 12). As computing soft-SPL requires rerunning all of our evaluations (itself slow for embodied AI tasks) we do not have enough computational resources to compute this in the rebuttal period. Therefore, we provide the soft-SPL by our AAP and EmbCLIP with $d^{*}_{m}=${$\pm0.05m, 0.2m$} in Fig. 13 in Appendix D.
>
> **W2. Overall, across all figures, it seems that the model is not affected by the rotation drifts. Even for the 180 degrees drift! The SR should necessarily decrease (slightly) when the agent is exposed to such high drifts.**
>
> This is because our two technical contributions, the action-impact embedding and order-invariant head, prevent the policy from remembering which action (i.e., $a^{i}$) would cause what specific effects (i.e., rotate $30^{\circ}$ with $\pm15^{\circ}$ drifts). More specifically, because our rotation actions cover all possible rotation degrees {$-150^{\circ}, -120^{\circ}, …, 0^{\circ}, 30^{\circ}, …, 150^{\circ}, 180^{\circ}$} with $\pm 15^{\circ}$ drifts, our policy observed all possible rotation outcomes within $[-180^\circ, 180^\circ]$. During the testing stage, taking 180 degree drifts as an example, our policy can recognize that the effect of $a^{i}$ (e.g. rotate$(30^{\circ}+180^{\circ}$ drifts$)$) is equivalent to the effect of another $a^{j}$ (e.g. rotate$(-150^{\circ})$) which was seen in the training stage. Likewise, since the movement magnitude {$0.05, 0.15, 0.25$} $\pm0.05$ can cover the range $[0, 0.3]$, our model performs well when the movement magnitude is within this range, but starts suffering with large drift $0.4$.

---

> > ### Author Response · Authors · 2022-11-18
> > **Response to Reviewer AsUJ**
> >
> > **W3. Figure 5 again, let's analyze one more time. For the ObjectNav experiment, it seems that the displacement drift does NOT affect the baselines, if the rotation drift is kept at 20º. Here there is no clear improvement with respect to the rest of the models. This aspect is worrying, as this experiment is more relevant than PointNav. Moreover, the result (of the baselines) is better the more drift, for models that have not been trained for it. These results are confusing, and in my opinion do not allow to offer a clear conclusion.**
> >
> > We hypothesize that baselines seem to perform relatively well during larger movement drift for the Object Navigation task is because the models have a better chance of quickly exploring the environment compared to when exposed to smaller movement drift. This fast exploration is highly favorable during Object Navigation as the agent has to “find” the object in the scene and must only be within $1.0$m of the object for it to be "found". For Point Navigation, on the other hand, the agent has to move to the target location within a pretty small distance ($0.2$m) which is quite challenging when experiencing large movement drifts. On the other hand, based on the rotation drifts, it is clear that our model performs considerably, and consistently, better than all baselines across all unseen drifts.

---

### Official Review · Reviewer_WkHK · 2022-10-25

**Confidence:** 4
**Correctness:** 3
**Technical Novelty And Significance:** 3
**Empirical Novelty And Significance:** 3
**Recommendation:** 6

**Clarity, Quality, Novelty And Reproducibility:**

The paper is easy to follow and the proposed solution is neat to improve the robustness of embodied agents. But why does the action-order encoder needs to generate the embedding for each $a^i$ where $i=0 \cdots n$? For any state change, it should only generate one embedding.

The idea proposed in the paper is similar to the action remapping idea (e.g. Edwards et al.) that associates latent actions (here the state changes) to the corresponding correct actions.
- Edwards et al. Imitating latent policies from observation. ICML, 2019.

The training parameters are available in the appendix for reproducibility. But I cannot find details about the network, e.g. the dimensions of the encoder output, in the paper. No other supplemental material is provided to help derive this information. It may be hard to reimplement the networks without this information.


**Strength And Weaknesses:**

Strength
- This paper proposes an approach that is useful to deal with action uncertainty in test time. This will add robustness to embedded agents.
- Compared to data augmentation which is arbitrary, this paper shows a systematic approach to deal with action drifts.

Weaknesses
- The approach assumes that the disabled actions can be remapped to one of the state changes the agent has seen before in training in order to take the right actions. For rotations, it is possible to cover all rotations in training, but for translation, it depends on the step size, for the translation that is really out-of-distribution state changes, this approach still cannot recover from that. This is shown as the lower success rate for the 0.4m drift. The authors need to make this assumption clear to the reader.
- One of the claims of the paper is that the embedding of state changes can help recover from action drift, but in the ablation study, LAC’s performance still drops when the drift increases which invalidates the claim of the paper. It is unintuitive why the action impact (i.e. state change) embedding doesn’t have an impact without OI.
- The ablation doesn’t really show OI is helpful as AAP uses a transformer head but LAC uses a linear head. The difference in performance can be the difference in model capacity. A fairer comparison will be a transformer head with positional encoding.


**Summary Of The Paper:**

This paper proposes an approach to learning action embeddings from state changes. For embedded agents who use these learned action embeddings, they do not rely on action semantics that overfits to the action labels. This paper also shows that an order-invariant (OI) transformer head can help choose actions rather than based on action semantics. Experiments on the AI-THOR environment show the proposed method is robust to several out-of-distribution action drifts and can quickly avoid using disabled actions.

**Summary Of The Review:**

This paper proposes a simple yet effective approach to improve robustness against action drifts. However, it misses the discussion of limitations of the proposed method. The description of the learned embedding and why an OI is needed is confusing.

---

> ### Author Response · Authors · 2022-11-18
> **Response to Reviewer WkHK**
>
> We thank you for the valuable feedback. In the following, we use W for Weakness and Q for Question.
>
> **W1. For rotations, it is possible to cover all rotations in training, but for translation, it depends on the step size for the translation that is really out-of-distribution state changes, this approach still cannot recover from that. This is shown as the lower success rate for the 0.4m drift. The authors need to make this assumption clear to the reader.**
>
> That’s correct. Our approach cannot cover all possible translations because the movement magnitudes are generally unbounded. We have updated the paper to include an example about how our AAP learns to deal with the unseen, but bounded, drifts in Appendix A to make it clear. Although in practice, typically robots do not have large drifts within a small time interval, we have some ideas to address this unbounded movement magnitude. One of them would be learning an agent which can augment its own action space during training. More specifically, the agent could learn to combine multiple move actions to form a new action with a large magnitude. For now, we leave this as an exciting future direction.
>
> **W2. It is unintuitive why the action impact (i.e. state change) embedding doesn’t have an impact without OI.**
>
> The proposed embedding of state changes has to be coupled with the OI-head to prevent the policy from overfitting to the action semantics. Intuitively: without the OI-head, the model gains little advantage from paying attention to the meaning of the impact embeddings during training because the order of these actions is fixed (i.e. if the first action is always MoveAhead during training, the information coming from the first action embedding is redundant). More technically, the action impact embedding fails without OI-head because the (order-aware) linear function in the Linear actor-critic has a fixed mapping between embeddings and each action. In this case, the policy still overfits to the action semantics. I.e., the policy “remembers” that action $a^{i}$ can produce, for example, only $30$ degree rotation with the linear actor-critic, while the embedding models the state changes. As a result, only having the action-impact embedding is not enough. The policy has to utilize the OI-head to prevent the actor-critic head remembering which action would cause what state changes.
>
> **W3. The ablation doesn’t really show OI is helpful as AAP uses a transformer head but LAC uses a linear head. The difference in performance can be the difference in model capacity. A fairer comparison will be a transformer head with positional encoding.**
>
> The suggested “a transformer head with positional encoding” is actually the “Action-Semantics” model shown in the blue color in Fig. 7. We replaced the Action-Impact Encoder by the Action-Order Encoder and the Action-Order Encoder is actually the positional encoding over the action space. The ablation studies show that both the Action-Impact Encoder and the OI-Head are necessary (green color). Using a Transformer Head with Positional Encoding cannot deal with unexpected drifts (blue color).
>
> **Q1. Why does the action-order encoder need to generate the embedding for each $a^{i}$
> where $i=0⋯n$? For any state change, it should only generate one embedding.**
>
> To model the impact resulting from different actions “independently”, our proposed method produces embedding for each action. In this way, the following policy network can make a decision based on which action could lead the agent to the best state. If the policy generates only one embedding for all actions, all action semantics would be mixed together into this single embedding and we have no chance to prevent the policy from overfitting to the action semantics only seen in the training stage.
>
> **Q2. The idea proposed in the paper is similar to the action remapping idea (e.g. Edwards et al.)**
>
> Thank you for the reference. We want to point out that our work is different from ILPO in the following ways: (1) Our framework does not have two stages of learning. We do not collect offline data to learn a latent forward network. Instead, we utilize the on-policy algorithm to train the policy with a forward prediction loss end-to-end. (2) Our testbed is in a visually complex and physical-enabled environment. It is pretty different from the environments used in the ILPO.
>
> **Q3. I cannot find details about the network, e.g. the dimensions of the encoder output, in the paper. No other supplemental material is provided to help derive this information. It may be hard to reimplement the networks without this information.**
>
> The dimension of the projected visual feature $r^*$ is $1024$ for PointNav and $3136$ for ObjectNav. The dimension of the goal embedding g is $32$. The hidden and the output embedding of the RNN are $512$-dim and $128$-dim. The Action-Order Enocder’s output dimension is $128$. We have updated our paper with these details in Appendix C. We will also release the code.

---

> > ### Comment · Reviewer_WkHK · 2022-12-05
> > **Thanks for the response!**
> >
> > I thank the authors for addressing most of my questions and concerns. Since this is an interesting paper and provides a clear experimental setup for the community to work on the problem, I'll update my review to leaning toward acceptance. However, the current proposed approach only works for bounded drifts, the experiment results in Fig 5 are still confusing as the model doesn't seem to be affected much by the movement drifts. It will be clearer if the paper can have plots to separate the effects of rotation and movement drifts.

---

### Official Review · Reviewer_DBZD · 2022-10-25

**Confidence:** 5
**Correctness:** 4
**Technical Novelty And Significance:** 4
**Empirical Novelty And Significance:** 3
**Recommendation:** 8

**Clarity, Quality, Novelty And Reproducibility:**

Clarity:
This is in general, a well written paper.

Quality:
The technical contribution is sound and suitable for ICLR.

Reproducibility:
'We will make the code and models for this work publicly available' - this declaration is to push SOTA - so will be reproducible then based on details given in paper body.

Novelty:
The enlisted contributions are indeed novel in this field of embodied AI (visual navigation tasks).

**Details Of Ethics Concerns:**

The docoloc plagiarism check result is 9% - so is compliant.

**Strength And Weaknesses:**

Strengths:
1. The experimentation section is strong to support most of their claims.
2. In their experiments, authors claim to outperform meta-learning approaches without requiring, computationally taxing, meta-learning training.
3. Out-of-distribution generalization (avoiding disabled actions)
4. For example, if the action a_i is Rotate(30◦) - this is a good example

Weakness:
1. The approach is not tested on (a) another sim environment like Gibson, MP3D AI Habitat (b) real robot - usually sim2real is not easy for this embodied AI tasks
2. It seems defective wheel plays a major role in errors. It can be due to surface change, sensor failure, environment noise. Some error modelling technique could have been pursued.
3. How is the learned RL model applicable to different robot type with different hardware sensors and motors? [A Limitation]
4. Embedding is not explained clearly - internals
5. The performance of the drift correction is tied up with the specific goal task at hand

**Summary Of The Paper:**

Authors propose Action Adaptive Policy (AAP) to adapt to unseen effects of actions during inference of an embodied AI navigation task. Experiments show that AAP is effective at adapting to unseen action outcomes at test time, outperforming a set of strong baselines by a significant margin, and works with disabled actions. The paper takes care of 2 cases of drifts - rotation and movement.

**Summary Of The Review:**

The paper content is a good contribution to SOTA.

Suggestions:
1. Make the abstract to the point - break large sentences into smaller form. Some result numbers and mention of PointGoal and ObjectGoal could also support overhaul.
2. action-stability assumption - this could be elaborated and can use a abbreviation later.
3. drift is a well known problem in SLAM community (loop closures) - how does this work leverage SOTA in those lines?
4. what is the time-space complexity of the pipeline at execution time given possible actions?
5. need some insights on how Fig 1. action commands of exact 30 degree rotate, 0.2 m translate passed
6. How the agent is learning should be explained philosophically, possibly by an example
7. Instead of SPL alone, other metrices can be tried out in future - soft-SPL, DTS, partial reward
8. we use DD-PPO - explain
9. What happens with fractional ground truth of rotation angles?


Miscellaneous:
1. Could have used line numbers in review for easy referral (i had to copy paste long lines)
2. Unlike work in training robust - grammar
3. General comment - break long sentences short for ease of reading
4. Make the contribution part writing simple
5. Fig. 2 should be elaborated more - choice of color...
6. expand abbreviations when used first
7. Choice of variable name / subscript - as input to produce o_0,t - does not look good and readability issue

---

> ### Author Response · Authors · 2022-11-18
> **Response to Reviewer DBZD**
>
> We thank you for the valuable feedback. In the following, we use W for Weakness and S for Suggestion.
>
> **W1. The approach is not tested in (a) another sim environment.**
>
> We added new experiments in a simple, 2D, modified multi-particle (MPE) environment from PettingZoo (Terry et al., 2020) on Point Navigation and Object Push tasks in Appendix F. For Point Navigation, the goal is to move the agent to a target location. For Object Push, the goal is to control the agent to push an object to a target location. The observation in this environment is the agent state and the target state. Actions in this environment accelerate the agent in a given direction. Please refer to Appendix F for more details about the environment. According to the results shown in Fig. 16, our proposed method outperforms all baselines consistently across unseen rotation drifts on both tasks.
>
> Regarding additional experiments in Habitat, we train the EmbCLIP and our AAP on Point Navigation task in Gibson v1. The simulator is Habitat-Lab v0.2.1. The training settings are the same as the settings used in AI2-THOR, including the same training drifts, same learning schedule, same optimization algorithm, and learning objective. During the evaluation, due to the time constraints and limited computational resources, we evaluate the model on only movement drift $d^{*}_m=0.2m$ and rotation drift $d^{\*}_r =$ {$\pm15^{\circ}, \pm30^{\circ}, \pm45^{\circ}, \pm90^{\circ}, \pm135^{\circ}, 180^{\circ}$}. The experiment details and results are included in Appendix G. As shown in Fig. 17, our AAP performs consistently across all rotation drifts. However, the EmbCLIP is struggling with larger rotation drifts.
>
> **W2. The defective wheel plays a major role in errors, but the error can be due to surface change, sensor failure, environment noise. Some error modeling techniques could have been pursued.**
>
> We learn to deal with unexpected action outcomes after applying an action. The unexpected outcomes could be caused by any of the mentioned sources. It is an interesting direction to learn which type of noise most likely causes the errors and react based on the learned model.
>
> **W3. How is the learned RL model applicable to different robot types with different hardware sensors and motors?**
>
> We believe the same model can be trained for other robot types if their sensory observations and actions fit the structure of our model. We expect that the gains seen will, however, be task-dependent. It is interesting future work to apply our approach to other types of embodiments.
>
> **W4. Embedding is not explained clearly - internals**
>
> The proposed embedding $e_{i, t}$ is an action-centric embedding of impact. Our goal is to model the state changes resulting from action $a^i$. More specifically, our action-impact encoder takes the previous visual feature, current visual feature, and the previous action $a^i$ as input. Then the RNN updates the state changes for action $a^i$ (from previous visual feature to the current visual feature resulting from the action $a^i$). Note that without the RNN, an agent that takes a MoveAhead action in front of a wall may erroneously believe that the move action does nothing. Finally, the output of the RNN will be registered into $e_{i, t}$ according to the $a^i$ index. As a result, each embedding $e_{i, t}$ will capture the outcomes produced by an action $a^i$.
>
> **W5. The performance of the drift correction is tied up with the specific goal task at hand**
>
> The drift correction is tied to the task. However, the approach is generalizable to other tasks if we provide the model with a reward signal corresponding to the tasks.
>
> **S1 & 2. Add numbers (results) to the abstract and action-stability assumption -> AS assumption.**
>
> We have updated our paper to include the major results in the abstract and use “AS assumption” as the abbreviation of “action-stability assumption” after we introduce it in the first paragraph in the introduction section.
>
> **S3. Drift is a well known problem in SLAM community (loop closures) - how does this work leverage SOTA in those lines?**
>
> We mostly focus on adaptation to unexpected action outcomes (for example, rotation with a broken wheel). Drift is one of the sources of noise that our model learns to handle. We believe our method can be combined with SLAM methods to provide better estimates for the state changes of the robot. We will explore this direction in the near future.

---

> > ### Author Response · Authors · 2022-11-18
> > **Response to Reviewer DBZD**
> >
> > **S4. What is the time-space complexity of the pipeline at execution time given possible actions?**
> >
> > We have updated Appendix B to include the information regarding the time-space complexity.
> > We provide a summary here: Theoretically, the time-complexity for the OI transformer head is O($n^2$), where $n$ is the number of actions in the action space. However, as the number of actions is usually fairly small, our AAP does not suffer from the same problems as transformer-based models with their $n^2$ complexity in the sequence length. At runtime, the time complexity depends on the machine (e.g., a server or a personal desktop). But in our evaluation desktop with a Intel i9-9900K CPU, 64G DDR4-3200 RAM, 2 Nvidia RTX 2080 Ti GPUs, the framework (w/ our AAP) spends $\approx$35 minutes evaluating $1.8$k val episodes with 5 parallel processes on the Point Navigation task. The average episode length is $\approx$117. As a result, the FPS (or interaction per second) is $\approx$100.3. During the training phase, we used an AWS machine with 48 vCPUs, 187G RAM, and 4 Nvidia Tesla T4 GPUs to train the policy. The FPS (or interaction per second) is $\approx$400 for our AAP. For the space, a single AI2THOR process uses 100M-200M GPU memory and our model occupies $\approx$2G GPU memory during the evaluation stage. Therefore, with 5 parallel processes at evaluation time, the framework takes $\approx$3G GPU memory.
> >
> > **S5. Need some insights on how Fig 1. action commands of exact $30$ degree rotate, $0.2$m translate passed.**
> >
> > $0.2$m translation or $30$ degree rotation corresponds to an action index in the action space. For example, $0.2$m translation corresponds to the $i$th action and $30$ degree rotation corresponds to the $j$th action. However, in this work, we do not directly embed the action index to an action embedding. Instead, we only use the action index to indicate which row (in Fig. 1) of the proposed action impact embedding should be updated. We have updated Fig. 1 to make it clear.
> >
> > **S6. How the agent is learning should be explained philosophically, possibly by an example.**
> >
> > Our two technical contributions, the action-impact embedding and order-invariant head, prevent the policy from remembering which action (i.e., $a^{i}$) would cause what specific effects (i.e., rotate $30^{\circ}$ with $\pm15^{\circ}$ drifts). More specifically, because our rotation actions cover all possible rotation degrees {$-150^{\circ}, -120^{\circ}, …, 0^{\circ}, 30^{\circ}, …, 150^{\circ}, 180^{\circ}$} with $\pm 15^{\circ}$ drifts, our policy observed all possible rotation outcomes within $[-180^\circ, 180^\circ]$. During the testing stage, taking 180 degree drift as an example, our policy can recognize that the effect of $a^{i}$ (e.g. rotate$(30^{\circ}+180^{\circ}$ drifts$)$) is equivalent to the effect of another $a^{j}$ (e.g. rotate$(-150^{\circ})$) which was seen in the training stage.
> >
> > **S7. other metrics can be tried out in future.**
> >
> > We have added the episode length, reward, distance to target, and soft-SPL to the Appendix D (Fig. 10, Fig. 11, Fig. 12, and Fig. 13).
> >
> > **S8. we use DD-PPO - explain.**
> >
> > The DD-PPO is the Decentralized Distributed Proximal Policy Optimization (Wijmans et al., 2020). It is a popular and standard model-free policy gradient algorithm. Embodied navigation works typically use DD-PPO to learn a policy since it provides relatively efficient training and stable convergence.
> >
> > **Miscellaneous.**
> >
> > We have updated our paper based on “Make the contribution part writing simple”, “Fig. 2 should be elaborated more - choice of color...”, and “expand abbreviations when used first”.

---

### Official Review · Reviewer_uyLJ · 2022-10-31

**Confidence:** 3
**Correctness:** 4
**Technical Novelty And Significance:** 3
**Empirical Novelty And Significance:** 4
**Recommendation:** 8

**Clarity, Quality, Novelty And Reproducibility:**

The paper is overall very well-written and easy to read. In particular, I appreciated that the authors took a great effort to motivate and explain well each of the architectural choices. The figures are also very clear and easily understandable on their own.

I'm not confident that the paper in its current form is reproducible. The authors committed to releasing the code, in the meantime, I suggest they include a detailed description of the architecture in the appendix (including a table of all hyperparameters and dimensions of the architecture). I also could not find a description of hyper-parameter sweeps or how well (and if) the hyper-parameters of baselines were tuned in these environments.

Here are some further suggestions for improving the clarity of the paper:
- explain the meaning of shaded area in Fig 5, 7 and 9
- add variance measures in Table 1

**Strength And Weaknesses:**

The reported results are very strong, the authors test the proposed method to both modest and severe drifts, as well with multiple disabled actions. They compare the proposed method to a variety of relevant baselines; including model-based RL, meta-learning, and sim2real methods; significantly outperforming all of them in almost all experiments, with a greater performance gap on bigger drifts.

The proposed model is very well-motivated and sensible, and I appreciated the detailed ablation experiments demonstrating the importance of both action embeddings and order-invariant action selection.

The main weakness of the paper is that the proposed method is evaluated on only one environment, AI2-THOR. While using embodied AI environment makes sense given the motivation for the proposed method is in robotics applications, I don't see a reason why this method wouldn't be more generally applicable in RL. The paper would be stronger with the addition of experiments on other environments, e.g. Mujoco, and some even simpler environments where the authors could more easily interpret the learned action embedding and policies, as well as shed light on the limitations of the method.

Some questions for the authors:
1. How many environment interactions does it typically take to learn action semantics in a new environment?
2. Do you have thoughts on how the proposed method could be adapted to continuous actions?
3. How exactly is action selecting $m_i$ and how is $m_i$ computed? I assumed you're selecting the last memory embedding transitions made under action $a_i$, hence only transitions made under action $i$ influence the embedding $e_i$; however it would be good to make this less ambiguous in the paper.
4. How well does the method perform in a non-stationary setting (i.e. where the action semantics change throughout the lifetime) or a changing distribution over actions (e.g. different drift sizes for each rotate action)? These questions are beyond the scope of the paper, but they seem like interesting application areas, and it would be good to get a sense of the method's limitations.

**Summary Of The Paper:**

The focus of this paper is the problem of generalization to changes in the effect of actions, which is referred to as action drift. To address this problem, the authors propose using: (a) action embeddings which are learned functions of transitions observed under that action, (b) performing action selection via an order-invariant transformer head. In the training phase, the model is trained on the distribution of action drifts and is then tested on drifts equal to or greater than those it has been trained on (without further adaptation of the model weights). The drifts explored here are modifications to either translation or rotation actions.

The proposed method is tested on the two tasks in AI2-THOR environment, where the proposed method significantly outperforms a variety of baseline methods, performing well even under OOD action drifts.

**Summary Of The Review:**

This paper proposed a method for fast adaptation to changes in the effect of actions during deployment. The effectiveness of the proposed method is well demonstrated in a series of experiments involving a variety of modifications to actions, strongly outperforming several relevant baselines.

---

> ### Author Response · Authors · 2022-11-18
> **Response to Reviewer uyLJ**
>
> We thank you for the valuable feedback. In the following, we use Q for Question, and S for Suggestion.
>
> **Q0. Addition of experiments on other environments, e.g. MuJoCo, and other simple environments. Shed light on limitations of the proposed approach.**
>
> We added new experiments in a simple, 2D, modified multi-particle (MPE) environment from PettingZoo (Terry et al., 2020) on Point Navigation and Object Push tasks in Appendix F. For Point Navigation, the goal is to move the agent to a target location. For Object Push, the goal is to control the agent to push an object to a target location. The observation in this environment is the agent state and the target state. Actions in this environment accelerate the agent in a given direction. Please refer to Appendix F for more details about the environment. According to the results shown in Fig. 16, our proposed method outperforms all baselines consistently across unseen rotation drifts on both tasks.
>
> While we have not yet had time to design environments that more fully demonstrate the limitations of our method, we sketch one such limitation below. As our method must explicitly take actions at the beginning of each episode to understand how they impact the environment, the agent must, unlike models with stronger action biases, spend more time exploring the effects of actions in the beginning of a new episode. This can result in suboptimal performance, for instance, the SPL evaluation shown in Fig. 9 of the Appendix D shows that our method obtains consistent SPL across all rotation drifts but the baselines achieves better results on seen rotation drifts (e.g., $\pm15^{\circ}$ seen in the training stage). While the cost of longer episode lengths (see Fig. 10 in Appendix D) caused by this exploration may seem like a small price to pay for the remarkable adaptability we have shown in our experiments.
>
> **Q1. How many environment interactions does it typically take to learn action semantics in a new environment?**
>
> To train a new policy on new tasks, the total number of training interactions can be found in Appendix B. In practice, we found that our method roughly achieves stable performance on the held-out validation set after 40M training interactions for Point Navigation and after 120M training interactions for Object Navigation.
>
> To quantify how many interactions AAP needs to adapt action semantics in a new environment, we compute the number of extra steps used by AAP compared to the baseline on point navigation task with seen training drifts (e.g., $d^{\*}_r=\pm15^{\circ}$ and $d^{\*}_m=±0.05m$). The reason to quantify the performance in this way is because we assume the baseline can achieve the most efficient route as it has seen that type of drift during training. Therefore, we can assume that the extra steps used by our AAP are the necessary steps to understand the action semantics in a new environment. The extra steps used by AAP is $5.97$ compared to EmbCLIP. Therefore, based on this assumption, our AAP typically needs $\approx6$ steps to understand the new action semantics and the corresponding drifts.

---

> > ### Author Response · Authors · 2022-11-18
> > **Response to Reviewer uyLJ**
> >
> > **Q2. Do you have thoughts on how the proposed method could be adapted to continuous actions?**
> >
> > Adapting our method to use with continuous actions is very interesting and an exciting direction for future work. Currently, we see two potential approaches:
> >
> > 1. Simple but perhaps less effective: convert continuous actions to (larger number of) discrete actions. For example, we might approximate a continuous navigation action space by using many rotation angles (e.g. {5, 10, 15 … 355}) and movement magnitudes (e.g. {0.025, 0.05, 0.075, … 0.025}). We have experimented with this approach and found that, while it does generally succeed, it generally results in lower sample efficiency in learning.
> >
> > 2. Explicitly model continuous actions: suppose an agent takes a $d$-dimensional continuous action $a_t\in R^d$ resulting in a state transition from $s_{t}$ to $s_{t+1}$. We can create an initial embedding of the effect of this action as usual by learning an embedding function $a_t = e_t = f(s_{t}, s_{t+1})$. There are then two questions: (a) as the agent will likely never take the same action $a_t$ twice (i.e. with 0 probability if using a continuous action distribution), how do we accumulate information about the effect of actions across time? (b) how do we design a "continuous" OI-head to use this information? An initial idea for (a) would be to treat the $(a_0, e_0), …, (a_t,e_t)$ pairs as a dataset with the $a_i$ representing the features and the $e_t$ representing the regression targets. We can then fit a nonparametric regression function $r_t$ to this dataset (e.g. k-nearest-neighbors) and use this to estimate the embedding $e$ corresponding to any potential action $a$ (i.e. $r_t(a) = e$). For (b), we would still use an OI-head as usual but would need some candidate actions $c^0, …, c^n \in R^d$ whose embeddings $r_t(c^0), …, r_t(c^n)$ would be fed into OI-head and then chosen among. There are many options as to how to sample $c^i$. The easiest would be to simply include all previously seen $a^{i}$ along with a few additional actions sampled in regions where the $r_t$ model is not confident (methods from active learning would be of interest here).
> >
> > **Q3. How exactly do particular actions correspond to particular memory vectors (i.e. when does action $a_t$ correspond to memory $m_i$?) How are the memory vectors computed? I assumed you're selecting the last memory embedding transitions made under action $a^{i}$, hence only transitions made under action $i$ influence the embedding $e_i$; however it would be good to make this less ambiguous in the paper.**
> >
> > Your understanding is correct. We use the action $i$ to index the last memory $m_i$ corresponding to action $i$ from a set of memories $M$. In this way, the transitions resulting from the action $i$ only have influence on the embedding $e_i$. We have updated our paper to clarify this point in Sec 4.2.
> >
> > **Q4. How well does the method perform in a non-stationary setting?**
> >
> > This is an interesting direction! Our current method is not designed to handle non-stationarity within episodes. We expect that the performance drops for non-stationary settings. To improve performance in non-stationary environments we likely require some mechanism by which to allow our agent to more rapidly "forget" its previous understanding of how actions impact the environment when it finds that said understanding no longer results in reliable predictions.

---

> > > ### Author Response · Authors · 2022-11-18
> > > **Response to Reviewer uyLJ**
> > >
> > > **S0: A detailed description of the architecture and hyper-parameters in the Appendix.**
> > >
> > > Thank you for the suggestion! We have updated our paper to include the details about model architecture in Appendix C. We summarize these details below. (a) Our model: we follow (Khandelwal et al., 2022) to set the dimension of the belief $b_t$ and hidden dimension to 512, and the output of ResNet is a 1568-dim goal-conditioned visual embedding. In the Action-Impact Encoder, the dimension of $r^*$ is 1024-dim for Point Navigation and 3136 for Object Navigation. The goal embedding is 8-dim, the hidden state of the GRU is 128-dim, and the action embedding is also 128-dim. In the policy network, the GRU is 512-dim. The OI-Head consists of 6 layers of Transformer Encoder with 16 heads, 512-dim hidden states, and 512-dim output. (b) EmbCLIP: the model architecture we use is the same one used in (Khandelwal et al., 2022) which is open-sourced on AllenAct (see https://github.com/allenai/allenact/blob/main/projects/objectnav_baselines/experiments/robothor/clip/objectnav_robothor_rgb_clipresnet50gru_ddppo.py). We plan to make a full open-source release of our codebase. (c) Meta-RL: the model architecture used here is the same as the one used in EmbCLIP, except that we add a MLP after the belief to predict the agent-state change for meta-learning. Instead of performing a meta-update every 6-steps, we perform meta-update every 20-steps, because the maximum allowed number of steps per episode was increased from 200 to 500 in our navigation tasks. The meta-update learning rate was set to 1e-4 (same as Wortsman et al., 2019). (d) RMA: the main model architecture is the same as the one used in EmbCLIP, except that we add a module to learn a latent code and an adaptation module to perform online adaptation. More specifically, we follow (Kumar et al., 2021) to encode the latent code $z$ (8-dim) using a 3 layer MLP (2->256->128->8) from motion and rotation drifts in the first training stage. In the adaptation stage, we follow (Kumar et al., 2021) to learn the adaptation module to predict a latent code from the past 16 agent states. In detail, the adaptation module consists of an MLP embedding (3->14->32->32) and a 3-layer 1D Conv ([32, 32, 8, 2]->[32, 32, 3, 1]->[32, 8, 3, 1]) over past 16 states. (e) Model-Based baseline: the model architecture is the same as our proposed model, except that the model produces action embeddings from the predicted state changes. In this way, this baseline can utilize the next state predictions associated with different actions for planning. In addition, it is important to note that the Policy Network for this baseline is the linear actor-critic, instead of the proposed OI head.
> > >
> > > The general learning hyperparameters, used across all the models, such as learning rate, iterations, optimizer, parameters for PPO, can be found in Appendix B.
> > >
> > > **S1. Explain the shaded area in Fig 5, 7, and 9.**
> > >
> > > The shaded areas show 1x standard deviation over different evaluations with $\pm$ motion drifts $d_m$ and $\pm$ rotation drifts $d_r$. For example, in Fig. 5, the evaluations of $0.05$m and $30^{\circ}$ includes the results produced by $(d_m, d_r) \in \{(0.05, 30^\circ), (-0.05, 30^\circ), (0.05, -30^\circ), (-0.05, -30◦)\}$. Because we consider +$d_r$ and -$d_r$ share similar semantics (likewise for +$d_m$ and -$d_m$), we show the average success rate with 1x standard deviation over them in the figures.
> > >
> > > **S2. Add variance measures in Table 1.**
> > >
> > > We plan to re-train all models with 2 additional and different random seeds and update the paper in the future. However, due to time constraints and resource limitations, we provide variances produced by our method and EmbCLIP here:
> > >
> > > - **Point Navigation**
> > > | SR (Avg. $\pm$ 1x std) | disable $\theta^{left}$  | disable $\theta^{right}$  |     Avg.    |
> > > |:---------------------:|:-----------------:|:------------------:|:--------------:|
> > > |     AAP (Ours)     |   $95.98 \pm 1.1$  |   $98.02 \pm 0.9$   | $96.99 \pm 1.4$ |
> > > |       EmbCLIP      |   $11.78 \pm 1.4$  |   $13.03 \pm 1.3$   | $12.57 \pm 1.6$ |
> > >
> > > - **Object Navigation**
> > > | SR (Avg. $\pm$ 1x std) | disable $\theta^{left}$  | disable $\theta^{right}$  |     Avg.    |
> > > |:---------------------:|:-----------------:|:------------------:|:--------------:|
> > > |     AAP (Ours)     |   $31.22 \pm 3.3$  |   $38.80 \pm 3.0$   | $35.00 \pm 4.9$ |
> > > |       EmbCLIP      |   $12.83 \pm 3.2$  |   $23.83 \pm 1.7$   | $18.33 \pm 6.1$ |

---

> > > > ### Comment · Reviewer_uyLJ · 2022-12-12
> > > > **Response to the authors**
> > > >
> > > > Thanks to the authors for a very detailed response!
> > > >
> > > > The authors have addressed my main concerns by including experiments on other domains (where the method still outperforms the baselines) and improving the clarity (by adding a more detailed description of the architecture and hyperparameters).
> > > >
> > > > I also appreciated the author's response regarding my questions about adapting the proposed method to continuous actions and non-stationary settings -- hope the former will make it to the discussion section of the paper.

---

### Author Response · Authors · 2022-11-18
**General Response**

We thank the reviewers for their valuable feedback. We are glad that reviewers _uyLJ_, _RBZD_, and _AaUJ_ recognize the effectiveness of our proposed method (SOTA results and great generalization ability). We also thank reviewers _uyLJ_ and _AaUJ_ for pointing out that our method is well-motivated and novel. Reviewers _uyLJ_ and _AaUJ_ also state that our detailed ablation studies are informative and demonstrate the importance of both action embeddings and Order-Invariant (OI) head. In addition, reviewer _WkHK_ mentions that our method is useful to action uncertainty at test time, and reviewer _AaUJ_ notes that our paper is well-written, clearly formulates the research problem, and contains informative figures.

After carefully reading all the reviews, we have added the following update:
- Follow suggestions from reviewers _uyLJ_ and _RBZD_ to revise the main paper to address the confusing points.
- In Appendix A, we provide an example of how our AAP learns to handle unseen drifts by using the Action-Impact Encoder and OI head.
- Provide more training and model details in Appendix B and C.
- Add more evaluation metrics, including episode length, reward, distance to target, soft-SPL, in Appendix D.
- Add the results of a new baseline inspired by (Nagabandi et al., 2018) in Appendix D.
- Add new experiments in a simple, 2D, modified multi-particle (MPE) environment from PettingZoo (Terry et al., 2020) on Point Navigation and Object Push tasks in Appendix F.
- Add new experiments in Habitat Point Navigation in Appendix G.

All changes are available in the revised pdf and highlighted in red color.

Please refer to our responses below for specific questions.

---

### Decision · Program_Chairs · 2023-01-20

**Decision:**

Accept: notable-top-5%

**Justification For Why Not Higher Score:**

N/A

**Justification For Why Not Lower Score:**

The method presented has several positive attributes that make it of wide interest to the community. It is well motivated; it stands to reason that paying attention how your actions changed your observations may improve your ability to adapt to changes in how those actions impact the environment. It clearly works well across datasets. And it could be applied to other problems, it is not restricted to RL for robotics. Finally, reviewers were unanimous in their support.

**Metareview: Summary, Strengths And Weaknesses:**

Summary: Enable robots to adapt to significant changes in their action space by embedding the impact of an action (the change in the observations caused by the action) and providing those embeddings when choosing the next action.
Strengths: A novel idea that is well motivated with clear experiments that demonstrate its value. Adapting to changes in the environment is of interest to the community as a whole and this method could see other uses. The authors addressed reviewer concerns, including new experiments such as running on Habitat.
Weaknesses: The method is limited in its ability to help with large drifts. It would be good if the manuscript was clearer about this and other limitations of the method.

**Note From Pc:**

if the above contains the word "oral" or "spotlight" please see: "oral" presentation means -> notable-top-5% and "spotlight" means -> notable-top-25%. As stated in our emails, we are disassociating presentation type from AC recommendations